# Trends in oxygenate/hydrocarbon selectivity for electrochemical $CO_{(2)}$ reduction to $C_2$ products

Hong-Jie Peng [1,2], Michael T. Tang[1,2], Joakim Halldin Stenlid [1,2], Xinyan Liu[2] & Frank Abild-Pedersen [1✉]

The electrochemical conversion of carbon di-/monoxide into commodity chemicals paves a way towards a sustainable society but it also presents one of the great challenges in catalysis. Herein, we present the trends in selectivity towards specific dicarbon oxygenate/hydrocarbon products from carbon monoxide reduction on transition metal catalysts, with special focus on copper. We unveil the distinctive role of electrolyte pH in tuning the dicarbon oxygenate/hydrocarbon selectivity. The understanding is based on density functional theory calculated energetics and microkinetic modeling. We identify the critical reaction steps determining selectivity and relate their transition state energies to two simple descriptors, the carbon and hydroxide binding strengths. The atomistic insight gained enables us to rationalize a number of experimental observations and provides avenues towards the design of selective electrocatalysts for liquid fuel production from carbon di-/monoxide.

[1] SUNCAT Center for Interface Science and Catalysis, SLAC National Accelerator Laboratory, Menlo Park, CA 94025, USA. [2] SUNCAT Center for Interface Science and Catalysis, Department of Chemical Engineering, Stanford University, Stanford, CA 94305, USA. ✉email: abild@slac.stanford.edu

Electrochemical reduction of carbon dioxide ($CO_2$R) into carbon-based chemicals and fuels offers an effective route to close the anthropogenic carbon cycle, store surplus renewable electricity, and build up a sustainable chemical industry[1,2]. Copper (Cu) and Cu-based materials are so far the only catalysts that produce high-value multicarbon ($C_{2+}$) species at a significant rate[3]. Among possible $C_2$ products, $C_2$ oxygenates (Oxys) such as ethanol and acetic acid are produced in liquid forms and possess high volumetric energy density and are compatible with existing infrastructure for easy storage and transportation[1,3]. The majority $C_2$ hydrocarbon (HC) product on Cu is ethylene, which is also a widely used chemical feedstock in industrial processes[3,4]. Despite the ability to produce $C_{2+}$ products, pristine Cu is not selective towards a specific $C_{2+}$ product. Thus, the fundamental selectivity issues of $CO_2$R on Cu is a challenge of great importance in catalysis and sustainable energy technologies. Substantial focus has been made in enhancing the selectivity of $C_2$ products for $CO_2$R[5–11].

A number of strategies have been developed to address the challenge of regulating $C_2$ Oxy/HC selectivity for $CO_2$R on Cu-based catalysts, including facet engineering[12,13], nanostructuring[14,15], oxide-derived Cu (OD-Cu)[6,11,16], alloying[9,17,18], and running $CO_2$R in tandem[7]. Specifically, direct reduction of carbon monoxide (COR) at alkaline conditions has been widely reported to yield significant $C_2$ Oxy species at more positive potentials than $CO_2$R[19]. This strategy was originally conceptualized on OD-Cu by Kanan and coworkers[6], and has later been validated for a broad range of Cu catalysts with distinct morphologies and under evaluation in various types of reactors[14–16,20,21]. In addition to intrinsic properties of Cu catalysts, such a universal phenomenon highlights the importance of extrinsic factors like potential and pH in steering $C_2$ Oxy/HC selectivity. The pH effects have been widely known for tuning the methane/$C_2$ selectivity[21–25]. However, despite the diverse reaction mechanisms proposed in theoretical works to understand the formation of ethylene and ethanol[26–31], the observed pH effects on $C_2$ Oxy/HC selectivity remain elusive. Model experiments on well-defined single-crystal Cu facets[12,32], as well as principle component analysis[33], provide statistical insight into product-specific active site motifs of Cu. Nevertheless, this insight is limited to pure Cu and as a consequence, the discovery of new materials is challenging. Thus, a fundamental look at the questions—*At which step does the last C–O bond break? And how does such a step correlate with potential and pH?*—is crucial for providing insight at the atomic-level and broaden our understanding when addressing the trends of $C_2$ Oxy/HC selectivity.

In this work, we investigate the reaction mechanism of COR towards major $C_2$ Oxy/HC products (*e.g.* ethylene, ethanol, and acetic acid) from reaction and activation energies calculated using density functional theory (DFT). Pathways trifurcating from a common intermediate, CHCO*, are shown to be responsible for generating specific $C_2$ Oxy/HC products. We observe that it is kinetically unfeasible to form ethylene through the OCHCH* and $CH_2CO*$ pathway. An electrochemical microkinetic model (exemplified on Cu(100)) is presented to adequately rationalize the experimental trends in $C_2$ Oxy/HC selectivity, revealing its dramatic dependency on electrolyte pH, applied potential, and the surface orientation of the Cu catalysts. We find that the distinct pH effect on the $C_2$ Oxy/HC selectivity stems from the selectivity-determining steps (SDSs) in each pathway having different proton-electron transfer (PET) numbers with water as the proton source. This insight allows us to propose two simple descriptors for the $C_2$ Oxy/HC selectivity on a given material - the adsorption free energies of C* and OH*. These energies serve as a measure of the carbophilicity and oxophilicity of the surface. The degree of carbophilicity is shown to primarily guide the $C_2$ Oxy/HC production. The two descriptors provide atomistic understanding of product-specific sites on Cu and analogous catalysts.

## Results and discussion

**Reaction pathways for COR towards various $C_2$ Oxy/HC.** Surface CO (CO*) is the key intermediate leading to further reduced products[3,22], and Cu(100) has been identified as the major exposed facet under reaction conditions[32,34]. We focus on COR and select the Cu(100) as the model surface for a detailed investigation of the reaction mechanism. Computational details and simplifications of the electrochemical models are shown in Supplementary Notes 1–5 with corresponding data and justification presented in Supplementary Figs 1, 2 and Supplementary Tables 1–8.

Figure 1 shows the most relevant reactions considered in this work. In a previous publication[35], we identified three major pathways for COR towards methane ($C_1$) and $C_{2(+)}$ products on Cu(100). The COH/C-H (grey) pathway leads to $C_1$ as the only product whereas the OCCOH (magenta) and COH/OC-C (black) pathways both result in significant $C_2$ production through a common intermediate dicarbon oxide (CCO*). A degree of rate control analysis shows that the total COR rate depends on energies of key surface species essentially present in the reaction pathways with a PET number ($n$, relative to CO) of less than two (Supplementary Fig. 3)[35]. In other words, reaction steps later in $n$ than CCO* only play a role in controlling the selectivity of $C_2$ Oxy/HC for COR. We therefore evaluate these later steps with a particular focus on different protonation steps. Neutral pH conditions for the bulk electrolyte (pH7) is considered throughout this section; however, the effect of pH will be discussed in a later section. In Fig. 2, we will show that while our energetics align closely with previous works by Calle-Vallejo et al.[26] and Cheng et al.[29], our barriers reveal allow for kinetic considerations regarding the formation of $C_2$ Oxy vs. HC on Cu surfaces.

*CCO*/CHCO*. Protons in solution can attack the surface-bound $\alpha$-C or the dangling carbonyl bonds to produce either CHCO* or CCHO*/CCOH*, respectively. The protonation of $\alpha$-C is shown to be thermodynamically more favorable and kinetically faster than the other two reaction steps, rendering CHCO* as the only possible protonation product of CCO* (Supplementary Fig. 4). Similar to CCO*, CHCO* has three possible reduction pathways, $CH_2CO*$, OCHCH*, or CHCOH* (Fig. 2a). Specifically, OCHCH formation is enabled through formation of a surface–O bond. Of the three branching intermediates OCHCH* is the most stable and CHCOH* the least (Supplementary Fig. 5a). Despite the poorer thermodynamic stability of CHCOH*, its formation exhibits the lowest $\Delta G_a$ of 0.65 eV at $U_{RHE} = 0$ V; whereas the other two steps have to overcome a free energy barrier at least 0.21 eV higher. Nevertheless, these differences in $\Delta G_a$ become smaller as the overpotential increases, due to the smaller charge-transfer coefficient ($\beta \sim 0.35$) for CHCO-H than for CHC-HO ($\beta \sim 0.45$) and OCCH-H ($\beta \sim 0.5$) (Supplementary Fig. 5b). Once the three species are generated on the surface, the subsequent reaction steps are shown to be facile at reducing conditions ($U_{RHE} = -0.73$ V; pH7) (Fig. 2b, d, and e). In each step trifurcating from CHCO*, the free energy of activation of the protonation reaction, $\Delta G_a$, controls the overall rate and hence the likely reduction pathway. Therefore, it is crucial to unveil at which step in each pathway the C–O bond breaks most easily and consequently identify the product specificity of the different pathways.

*The CHCOH pathway.* Among the three possible protonation steps, the protonation of the hydroxyl group in CHCOH* to produce water and CCH* is the most kinetically favorable pathway (Fig. 2f and Supplementary Fig. 6). Cheng et al. also suggested the predominance of CCH formation through protonation[29]. And the extremely low coverage of H* ($\sim 10^{-8}$ monolayer) on Cu(100) strongly limits the CHCHOH formation

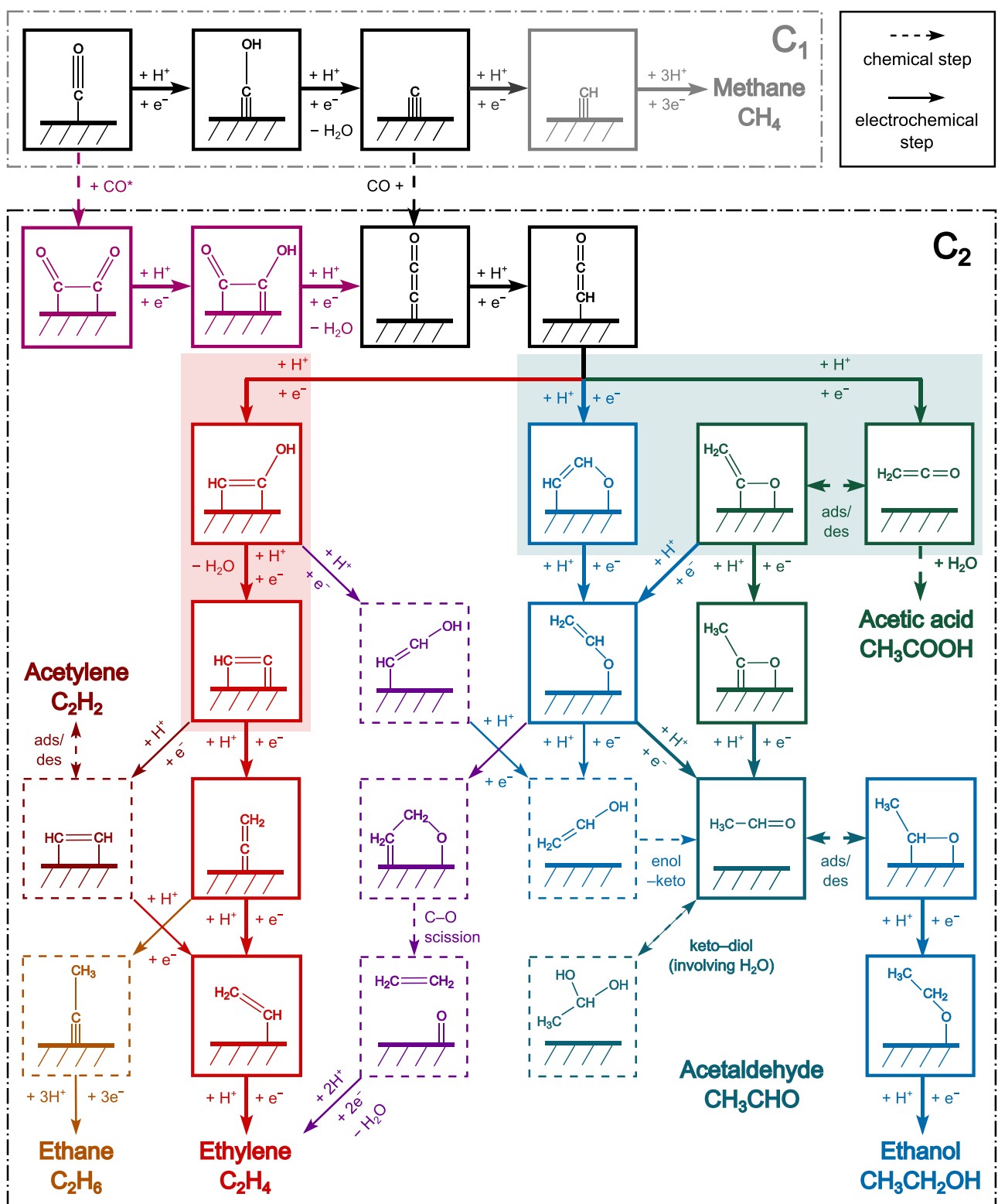

**Fig. 1 Schematic diagram of reaction steps beyond CO.** Reaction intermediates involved in major pathways are shown in solid squares with different colors: (1) Common pathways via COH*/C* towards both $C_1$ and $C_2$ products (black/grey) and additional $C_2$ pathway via OCCO*/OCCOH* (magenta). The above pathways control the total COR rate. (2) Major selectivity-determining pathways trifurcating from CHCO*, including the CHCOH pathway towards ethylene as the main $C_2$ HC product (red), the OCHCH pathway towards acetaldehyde and ethanol (blue), and the $CH_2CO$ pathway towards all three major Oxy species, i.e., acetic acid/acetate, acetaldehyde, and ethanol (green). The important steps governing whether the last C–O bond is cleaved or preserved are marked with light red and light cyan backgrounds, respectively. (3) Minor pathways branching off the major pathways are indicated by dashed squares, including the CHCOH pathway towards minor $C_2$ HC products (brown for acetylene and gold for ethane), the CHCOH pathway towards $C_2$ Oxys (purple, via CHCHOH*), and the OCHCH/$CH_2CO$ pathways towards ethylene (purple, via $OCH_2CH_2$*). Solid arrows refer to electrochemical protonation steps and dashed arrows refer to chemical processes.

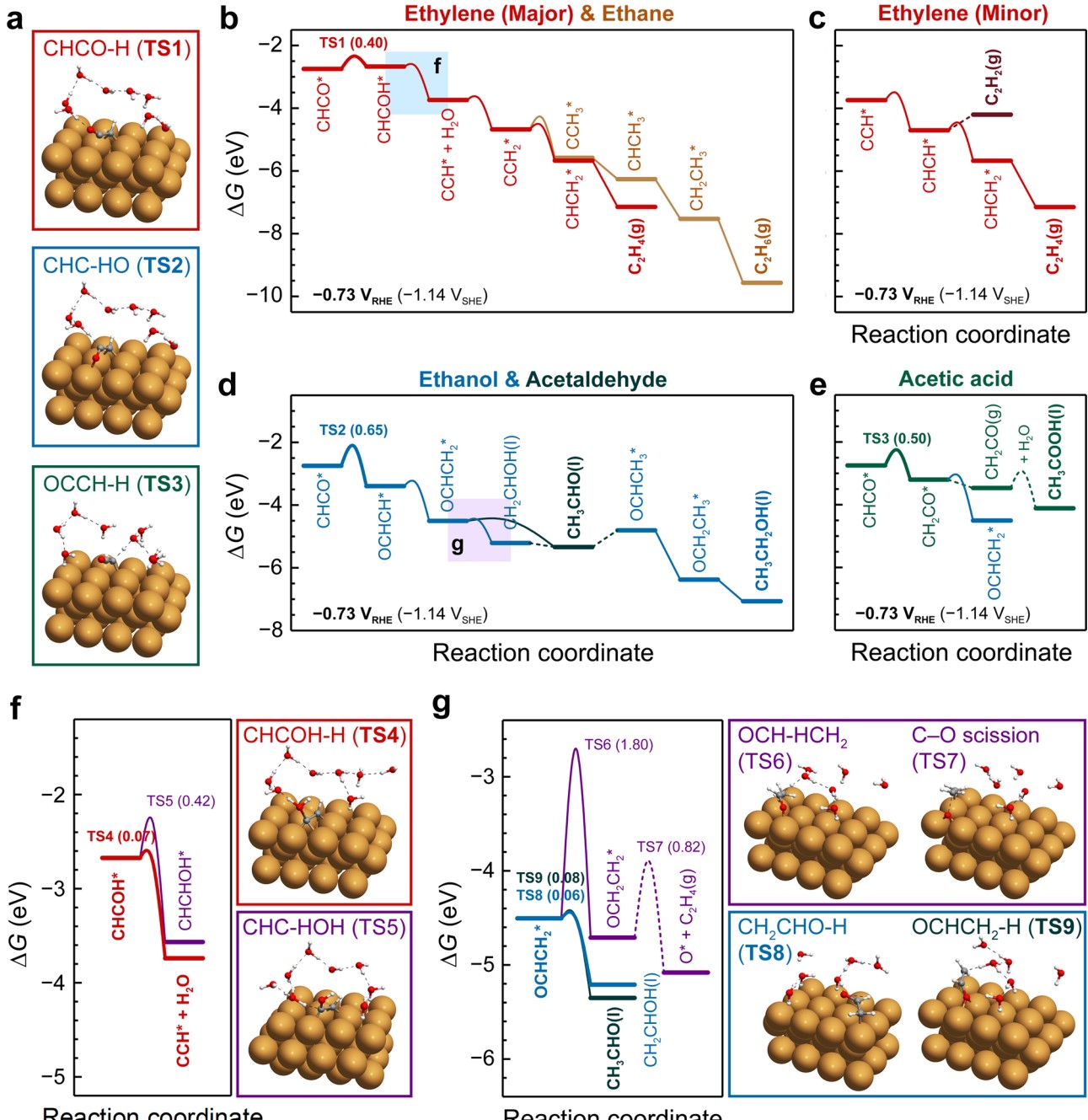

**Fig. 2 Free energy diagrams (FEDs) of COR on Cu(100) at $U_{RHE} = -0.73$ V (pH7). a** TS structures of CHCO-H (TS1), CHC-HO (TS2), and OCCH-H (TS3) protonation. FEDs showing **b** the major CHCOH pathway towards ethylene (red) and ethane (yellow), **c** the minor CHCOH pathway towards ethylene (red) and acetylene (brown), **d** the OCHCH pathway towards ethanol (blue) and acetaldehyde (navy), and **e** the $CH_2CO$ pathway towards acetic acid (green) and an intermediate (blue) leading to the other two $C_2$ Oxy products. The colored squares in **b** and **d** highlight regions where key competing reaction steps appear from branching pathways as illustrated in Fig. 1: **f** CHCHOH formation from CHCOH* (purple), possibly leading to $C_2$ Oxys and **g** ethylene formation from $OCHCH_2$* (purple) as proposed by Calle-Vallejo et al.[26] and based only on the thermodynamic stabilities of surface species. Important TS structures, CHCOH-H (TS4) and CHC-HOH (TS5) protonation are shown in **f**, and OCH-$HCH_2$ (TS6), $CH_2CHO$-H (TS8), and $OCHCH_2$-H (TS9) protonation, as well as C–O scission of $OCH_2CH_2$* (TS7) are shown in **g**. Electrochemical protonation steps are shown as solid lines while chemical processes are shown as dashed lines. All energies in the FEDs are referenced to CO(g), $H_2$(g), and $H_2O$ based on the computational hydrogen electrode (CHE) model. The values in parentheses in **b–g** represent the $\Delta G_a$ at $U_{RHE} = -0.73$ V (pH7). Source data are provided as a Source Data file.

via surface hydrogenation[35]. Since the C–O bond breaks easily through CHCOH-H protonation, the CHCOH pathway is consequently identified to exclusively produce $C_2$ HCs. Once the CCH* is formed, a series of $CH_xCH_{y+1}$ * intermediates ($x, y > 0$) follows, among which CCH* → $CCH_2$* → $CHCH_2$* → $C_2H_4$(g) is shown to be the major pathway towards ethylene (Fig. 2b, c). The

absence of $C_2$ HC products, acetylene and ethane, observed experimentally may be attributed to the slower H-CCH protonation and $CCH_2$-H protonation steps compared to the steps towards ethylene, i.e. CCH-H and H-$CCH_2$, respectively. In addition, the relatively strong binding strength of HCCH* limits the desorption of acetylene and hence favors subsequent reduction.

*The OCHCH pathway.* Because OCHCH* is thermodynamically more stable than CHCOH* and CH$_2$CO*, it is typically considered an inevitable intermediate in several proposed reaction mechanisms of ethylene formation[26,27]. As proposed by Calle-Vallejo et al.[26], one of the reaction steps involves protonation of OCHCH$_2$*, the most stable intermediate in the OCHCH* reduction on Cu(100), and simultaneous C–O bond scission to generate atomic oxygen (O*) and the release of C$_2$H$_4$. Here we observe that OCH$_2$CH$_2$* is a stable intermediate along the reaction coordinates between two TSs, thus demonstrating the disjointed nature of the reaction OCHCH$_2$* + H$^+$ + e$^-$ → O* + C$_2$H$_4$(g) (Supplementary Fig. 7). In sharp contrast to the extremely high barriers of directly producing ethylene from OCHCH$_2$*, reducing OCHCH$_2$* to either acetaldehyde or vinyl alcohol is almost spontaneous at reducing conditions (Fig. 2g). Vinyl alcohol is unstable in an aqueous solution at room temperature and ultimately isomerizes to acetaldehyde[36]. Once acetaldehyde is formed, ethanol has been experimentally validated as the only further reduced C$_2$ product[22,37]. Compared to OCHCH$_2$*, OCH$_2$CH* and CHCHOH* as two other possible intermediates through reduction of OCHCH* are demonstrated to be less favorable (Supplementary Fig. 8). The above analysis suggests that the possible C$_2$ products from the OCHCH pathway are limited to acetaldehyde and ethanol (Fig. 2d).

*The CH$_2$CO pathway.* Adsorbed CH$_2$CO* tends to detach from the surface and generate molecular ethenone in solution. Owing to the highly reactive nature of cumulative double bonds, ethenone is susceptible to nucleophilic attack from water to generate acetic acid, i.e., acetate at neutral and alkaline conditions (Fig. 2e). The hydration barrier was experimentally measured to be ~0.6 eV[38], which is in accordance with simulations performed by Luc et al.[8]. Thus, the CH$_2$CO pathway naturally contributes to the acetate generation. Stronger nucleophilicity of hydroxide anions (OH$^-$) than water further favors the reaction between ethenone and OH$^-$ under high alkaline conditions[6,8,16]. The incorporation of solvent oxygen from either water or OH$^-$ is also supported by $^{18}$O isotope labeling experiments[8,16]. Once acetate is formed, the electrostatic repulsion between the negatively charged electrode and this anion species, will prevent the acetate from further reduction. Reducing CH$_2$CO* toward acetaldehyde/ethanol is the relevant process competing with acetate formation, which, however, has hardly been investigated[28]. Protonation of CH$_2$CO* results in surface species CH$_3$CO*, OCHCH$_2$*, and CH$_2$COH*, respectively, all of which tends to be further reduced to C$_2$ Oxys (Supplementary Figs. 9 and 10). Compared to the OCHCH pathway, the only difference lies in the ability of CH$_2$CO pathway to produce acetate.

From this we conclude, that the production of major C$_2$ Oxy/HC species under CO$_{(2)}$R conditions stems from a common intermediate of CHCO*. The protonation steps of CHCO* play a vital role in determining the C$_2$ product selectivity. The CHCOH pathway exclusively results in C$_2$ HC, whereas the OCHCH and CH$_2$CO pathways lead to C$_2$ Oxy only.

**Microkinetic model of COR and pH effects on C$_2$ Oxy/HC selectivity.** To further illustrate the important role of CHCO protonation steps for C$_2$ Oxy/HC products, we have developed a mean-field microkinetic model that accounts for adsorbate-adsorbate interactions (see details in Supplementary Note 6)[25]. Given the intrinsic DFT errors (±0.15 eV) on barriers and intermediate energies and the uncertainties brought by the parameterization and the varying solvent structures, the microkinetic model only serves as a tool for qualitative comparison with experimental trends in activity and selectivity.

Figure 3a shows the trends in theoretical COR current densities on Cu(100) obtained through microkinetic modeling, which are directly comparable with the experimentally measured CO$_{(2)}$R current densities on planar polycrystalline Cu (pcCu) shown in Fig. 3b. Two pH conditions were considered, neutral (pH7) and alkaline (pH13). Despite the discrepancies in the absolute magnitudes, our model accurately predicts the potential-dependent variations in activity and selectivity and reproduces several key characteristics in experimental observations:

- When compared to methane, larger shifts in onset potentials are observed for all C$_2$ species by varying the pH. This can be attributed to the difference in PET number (relative to CO) for the rate-determining steps (RDSs) of dominant C$_2$ and C$_1$ pathways at low overpotentials, i.e., OCCO-H protonation for C$_2$ and COH-H protonation for C$_1$. This insight has been described in detail in our previous work[35].
- Ethylene is a more abundant C$_2$ product than ethanol and acetate. The predicted rate of ethylene relative to ethanol/acetate on Cu(100) is even higher than that experimentally obtained on pcCu, which is in agreement with observations by Hori et al.[12].
- In the low-current-density kinetically-controlled region and at pH13, the potential dependencies of ethylene, ethanol, and acetate are slightly different, indicated by a steeper Tafel slope for ethylene than for ethanol and acetate. This feature originates from the competition between multiple reaction pathways (e.g., the three parallel pathways (CHCOH, OCHCH, and CH$_2$CO) for CHCO*, the ethenone hydration *vs.* CH$_2$CO reduction for CH$_2$CO*) and the fact that these competing pathways possess distinct potential dependencies. We will discuss the potential and pH dependencies in more details later.

We also note that the larger discrepancy in predicting acetate production at alkaline condition could be attributed to the additional pathway of nucleophilic reaction between OH$^-$ and ethenone or surface carbonyl compounds such as CCO* and CHCO*[39]. Furthermore, studies that attempt to determine of the Tafel slopes for CO$_{(2)}$R are often based on the RDS—through $-\frac{2.3k_B T}{e(n+\beta)}$, where $k_B$ is the Boltzmann constant, $T$ the temperature, $e$ the elementary charge, $n$ the PET number before the RDS, and $\beta$ the charge-transfer coefficient of the RDS[25,40]—we note that the RDS approach of determining Tafel slopes is valid only for analyzing the total rate of CO/CO$_2$ consumption or C$_1$/C$_2$ formation. To further analyze the formation of specific C$_2$ products derived from the same intermediate formed after the RDS, the Tafel slope of each product consists of two individual components: one from the total C$_2$ formation and the other from relative rates between different pathways. Due to the dynamic nature of the second component, one should be cautious with corresponding Tafel analysis.

Besides the variation of reaction rates with potential, pH is seen to have a profound effect on the C$_2$ Oxy/HC selectivity (Fig. 3c). To prevent the interference from the different PET number per product molecule, we employed C$_2$ Oxy/HC molecular ratios rather than FEs for the selectivity analysis. Experimental values were adopted from the literature where a broad range of Cu catalysts and reactor types were investigated. We classify the Cu catalysts into pcCu electrodes[4,6,20,21,41], OD-Cu electrodes[6,16,41], single-crystal Cu electrodes[13], and high-roughness-factor (RF) Cu electrodes (mostly derived from oxide)[14,15]. For a fair comparison, we only considered tabulated literature values with product distribution and CO$_2$R in 0.1 M KHCO$_3$ electrolyte or COR in 0.1 M KOH electrolyte for Fig. 3c. The 0.1 M cation concentration

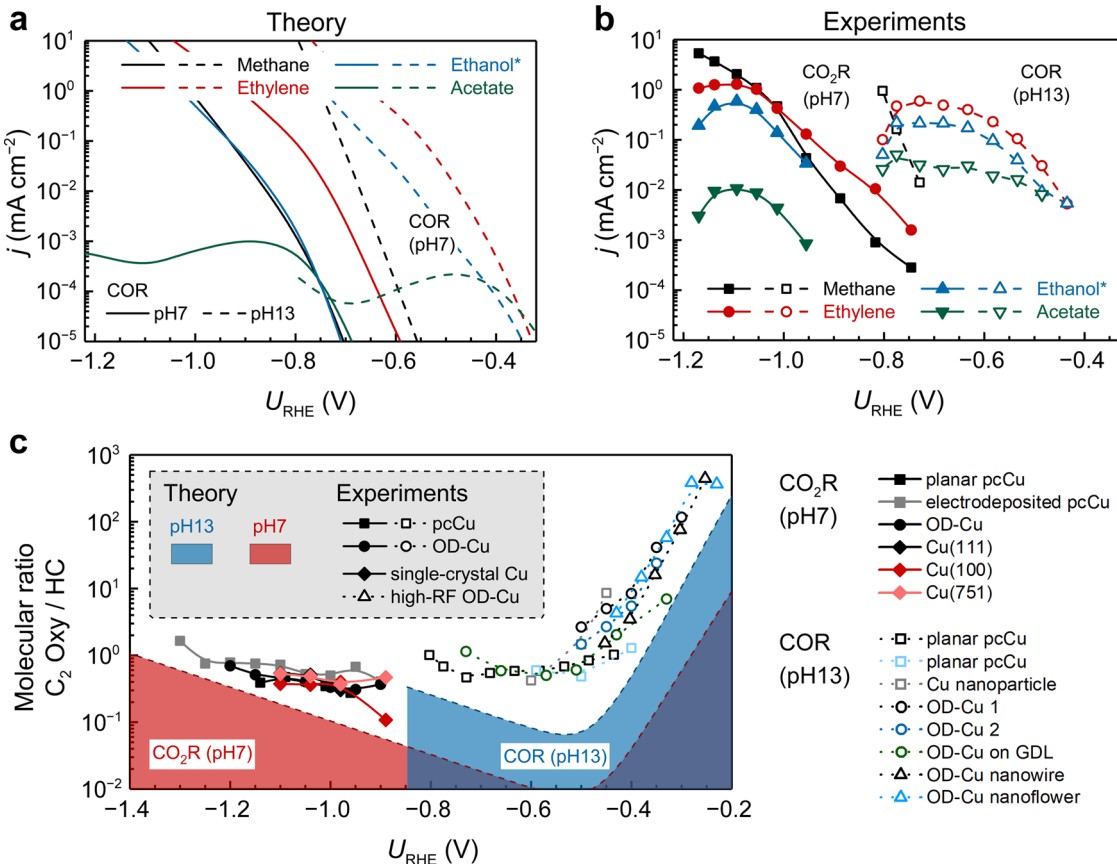

**Fig. 3 Trends in CO$_{(2)}$R current densities ($j$) and C$_2$ Oxy/HC molecular ratios with varying potentials and pH. a** Theoretical COR and **b** experimental CO$_{(2)}$R polarization curves. The experimental data for CO$_2$R (in 0.1 M KHCO$_3$, approximately pH7) and COR (0.1 M KOH, pH13) are obtained on planar pcCu from ref. [4] and ref. [21], respectively. The current densities for CO$_2$R were intentionally converted by considering the difference in electron transfer number per product molecule between CO$_2$ and CO as the initial reactants (see details in Supplementary Table 9). The current density of ethanol shown herein is the sum of ethanol and acetaldehyde and denoted with asterisk (*). **c** C$_2$ Oxy/HC molecular ratios for CO$_2$R (pH7) and COR (pH13) as a function of $U_{RHE}$ on a broad range of Cu-based catalysts reported in the literature. Solid and hollow plots correspond to data from CO$_2$R (pH7) and COR (pH13), with solid and dotted lines to indicate the trend, respectively. Dashed lines in red and blue show the theoretically predicted trends in C$_2$ Oxy/HC molecular ratios at pH7 and pH13, respectively. Data from literatures: (CO$_2$R pH7) planar pcCu;[4] electrodeposited pcCu and OD-Cu;[41] single-crystal Cu(111), Cu(100), and Cu(751);[13] (COR pH13) planar pcCu (black);[21] planar pcCu (light blue);[20] Cu nanoparticle, OD-Cu 1, and OD-Cu 2;[6] OD-Cu on gas diffusion layer (GDL);[16] OD-Cu nanowire;[14] OD-Cu nanoflower[15]. All the data for **c** are listed in Supplementary Table 10. Source data are provided as a Source Data file.

was selected to be consistent with the parameters in our electric field model; *albeit* we show that variation of the model parameters has little effect on the overall trend (Supplementary Fig. 11). It is clearly observed that the C$_2$ Oxy/HC molecular ratios decrease drastically when $U_{RHE}$ decreases to approximately −0.5 V, which is then followed by a slight increase in the more negative potential region (−0.5 V to −0.85 V). Our model accurately predicts such an inverse Volcano-like characteristic. The observed discrepancy in absolute magnitudes could be attributed to the absence of undercoordinated sites on Cu(100) in our simulation, whereas undercoordinated sites present in defect- or grain-boundary-rich pcCu or OD-Cu have been suggested as C$_2$ Oxy-specific active sites in previous studies[6,13,33]. We will unravel the atomistic origin of C$_2$ Oxy-specific activity on these sites in the last section.

In general, we ascribe the experimentally observed difference between CO$_2$R and COR in Fig. 3c to a pH effect and the different operating windows of $U_{RHE}$. The very high C$_2$ Oxy selectivity of alkaline COR cannot be reasoned solely through a promoted acetate formation with increasing OH$^-$ concentration[6,8,16]. In fact, the ethanol/ethylene molecular ratios (with acetate

formation ignored) also exhibit similar trends, suggesting a common factor underpinning this pH effect (Supplementary Fig. 12). To explore this common factor, we conducted a degree of selectivity control (DSC) analysis—a mathematical approach that explicitly links the product sensitivity to certain rate-determining reaction intermediates (see details in Supplementary Note 7). Figure 4 depicts the DSCs for ethylene, ethanol, and acetic acid as a function of potential and pH. At pH7, Cu(100) is found to be extremely selective towards ethylene, and the ethylene production is insensitive to any of the energy states in a range of $U_{RHE}$ from approximately −0.4 V to −0.8 V; only when $U_{RHE}$ decreases further, the difference between CHCO-H$^{TS}$ and OCCH-H$^{TS}$ becomes smaller and the CH$_2$CO pathway starts to compete with the CHCOH pathway (Fig. 4a). The formation of C$_2$ Oxy shown in Fig. 4b, c is strongly limited by the enhanced stabilization of CHCO-H$^{TS}$ (a DSC close to −1) on Cu(100). A stabilization of either CHC-HO$^{TS}$ and OCCH-H$^{TS}$ with a DSC close to 1 should favor ethanol and acetic acid formation in the −0.4 V to −0.8 V $U_{RHE}$ potential range.

Besides the three TSs for the initial protonation of CHCO*, we also note that the relative rate of the CHCOH pathway is

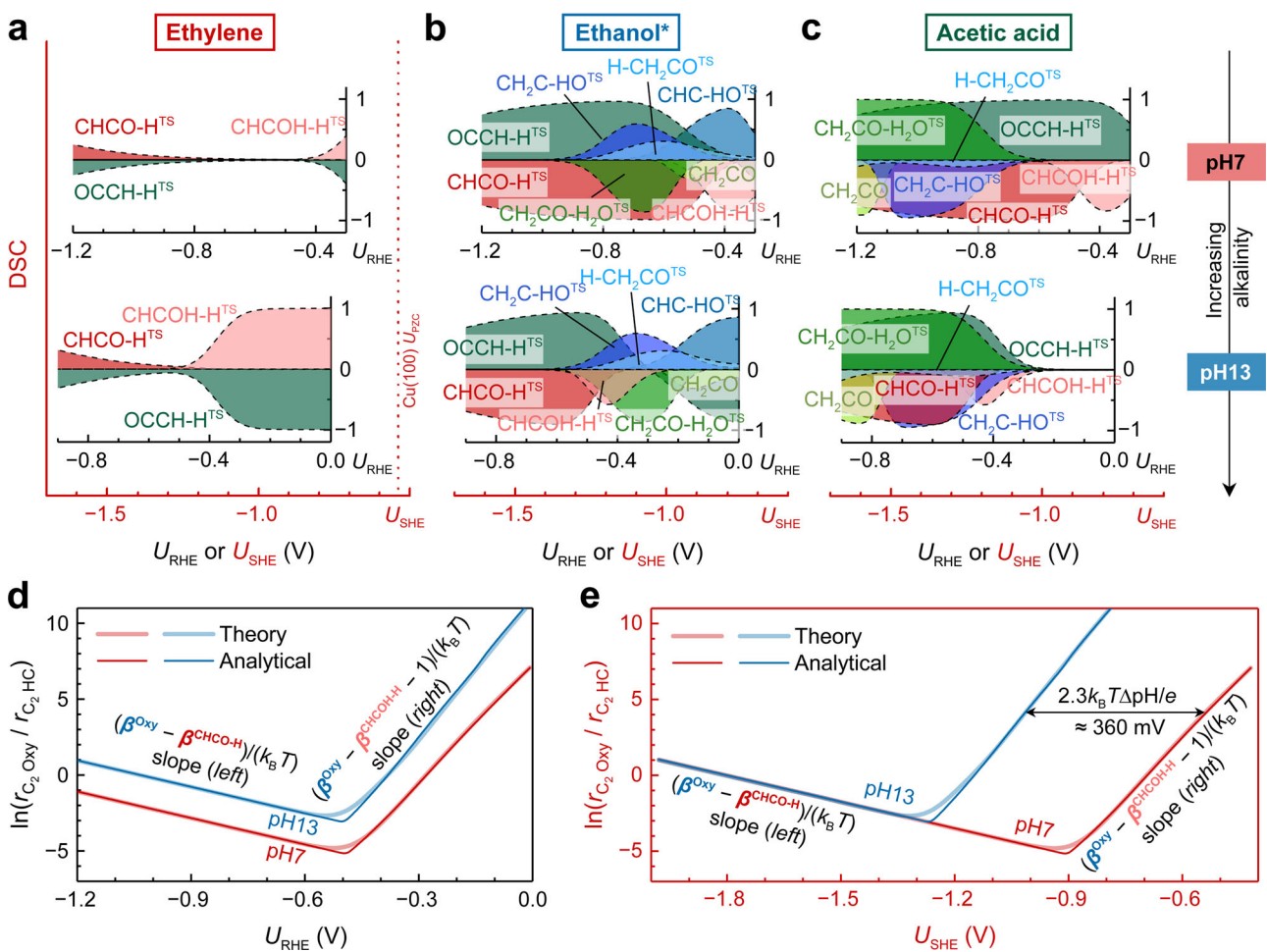

**Fig. 4 Origin of the pH effects on the $C_2$ Oxy/HC selectivity for COR.** Analyses of DSC for **a** ethylene, **b** ethanol + acetaldehyde (denoted as ethanol*), and **c** acetic acid. Similar to the degree of rate control conceptualized by Campbell et al.[60], DSC serves as a powerful tool to quantify the magnitude of selectivity controlling by a certain intermediate or a TS. A positive (negative) value of DSC indicates that the corresponding reaction intermediate or TS needs to be stabilized (destabilized) in order to enhance the selectivity. The boundary values of 1 and −1 represent full selectivity control by the intermediates. Only the energy states having significant contribution to the DSC, i.e. absolute value of DSC > 0.01, are shown. Key intermediates and TSs associated with the CHCOH, OCHCH, and $CH_2CO$ pathways are shown in reddish, bluish, and greenish colors, respectively. From the upper to lower panels, the pH of bulk electrolyte increases from 7 to 13. Analytical approximation of $\ln(\frac{r_{C_2Oxy}}{r_{C_2HC}})$ as a function of **d** $U_{RHE}$ and **e** $U_{SHE}$ at pH7 and pH13, in comparison with the theoretical results obtained through microkinetic modeling. The slopes of the two legs are given by Eq. (1) and Eq. (2), respectively. $\beta^{Oxy}$ could be either $\beta^{OCCH-H}$ or $\beta^{CHC-HO}$ depending on the dominant $C_2$ Oxy formation pathway. Here $\beta^{OCCH-H}$ was employed for Cu(100). Source data are provided as a Source Data file.

controlled primarily by the second protonation step (CHCOH-$H^{TS}$) at low overpotentials ($U_{RHE} > -0.5$ V). This factor becomes more predominant for ethylene formation at pH13, where CHCOH-$H^{TS}$ exhibits a significantly higher DSC than at pH7 when $U_{RHE} > -0.4$ V. Such a shift with pH in the SDS is analogous to the shift in the RDS for the COH pathway to produce methane (Supplementary Fig. 3)[35]. Because of the additional protonation step required to produce CCH* + $H_2O$ from CHCOH*, increasing pH suppresses the CHCOH pathway at the same absolute potential (*vs.* standard hydrogen electrode, $U_{SHE}$) and consequently yields higher $C_2$ Oxy/HC selectivity at low overpotentials (Supplementary Fig. 13). This shift also rationalizes the inverse Volcano shown in Fig. 3c. On the right leg ($U_{RHE} > -0.5$ V), the SDS for $C_2$ Oxys (mainly OCCH-$H^{TS}$ on Cu(100)) competes with CHCOH-$H^{TS}$; while on the left leg ($U_{RHE} < -0.5$ V), the SDS for $C_2$ Oxys competes with CHCO-$H^{TS}$. With this insight, we can derive a simple analytic expression for the $C_2$ Oxy/HC molecular ratio when the $CH_2CO$ pathway is more dominant than the OCHCH pathway for $C_2$ Oxy formation (see details in Supplementary Note 8).

On the right leg we have $\Delta G_a^{CHCOH-H} > \Delta G_a^{CHCO-H}$, such that

$$\ln\left(\frac{r_{C_2Oxy}}{r_{C_2HC}}\right) \approx \frac{(\beta^{OCCH-H} - \beta^{CHCOH-H} - 1)eU_{RHE} + (\Delta G_{a,0}^{CHCOH-H} - \Delta G_{a,0}^{OCCH-H})}{k_B T}$$

(1)

whereas on the left leg we find $\Delta G_a^{CHCOH-H} < \Delta G_a^{CHCO-H}$, such that

$$\ln\left(\frac{r_{C_2Oxy}}{r_{C_2HC}}\right) \approx \frac{(\beta^{OCCH-H} - \beta^{CHCO-H})eU_{RHE} + (\Delta G_{a,0}^{CHCO-H} - \Delta G_{a,0}^{OCCH-H})}{k_B T}$$

(2)

Here $r_{C_2Oxy}$ and $r_{C_2HC}$ are the rate of $C_2$ Oxy and HC formation, respectively, $\Delta G_{a,0}$ the activation energy relative to CHCO* at $U_{RHE} = 0$. With this we can quantify the slope and intercepts of both legs. The potential and pH dependence of key intermediate energies present in Eq. (1) and Eq. (2) induces the upward shift of the inverse volcano with increasing pH on a $U_{RHE}$ scale or, *videlicet*, the shift of the right leg of the volcano towards more negative potentials on a $U_{SHE}$ scale (Fig. 4d, e). Similarly,

increasing the local pH also favors the $C_2$ Oxy formation, which is in agreement with experiments performed on high-RF Cu electrodes under neutral conditions (Supplementary Fig. 14)[34,41].

In addition to the key energies determining the $C_2$ Oxy/HC selectivity, Fig. 4b, c also reveal the competing processes governing the ethanol/acetate selectivity. Reduction steps of $CH_2CO^*$, i.e. H-$CH_2CO$ and $CH_2C$-HO, possess a positive DSC for ethanol production but a negative DSC for acetate; in contrast, the hydration barrier of molecular ethenone exhibits the opposite trend. Especially at pH13, the acetate formation is insensitive to any of the present energies when $U_{RHE} > -0.3$ V, suggesting the acetate as the most likely $C_2$ product at small overpotentials. This is in excellent agreement with the experimental observation by Li et al[6]. Regardless, these energies have no effect on tuning the overall $C_2$ Oxy/HC selectivity according to Fig. 4a.

**Selectivity maps with $\Delta G_{C^*}$ and $\Delta G_{OH^*}$ as descriptors.** The above computed energetics, microkinetic modeling, and the DSC analysis allow us to identify three key reaction steps that determines the $C_2$ Oxy/HC selectivity:

$$CHCO^* + H^+ + e^- \rightarrow CHCOH^* \text{ (CHCO-H protonation)} \quad \text{(i)}$$

$$CHCO^* + H^+ + e^- \rightarrow OCHCH^* \text{ (CHC-HO protonation)} \quad \text{(ii)}$$

$$CHCO^* + H^+ + e^- \rightarrow CH_2CO^* \text{ (or } CH_2CO(g) + *)$$
$$\text{(OCCH-H protonation)} \quad \text{(iii)}$$

A theoretical description of the $C_2$ Oxy/HC selectivity turns into a simple scheme where TS energies of these key steps define the reaction products. Utilizing that these energies can be adequately described through scaling by $\Delta G_{C^*}$ and $\Delta G_{OH^*}$, we have approximated the activation energies of the three protonation reactions on close-packed metal (100) surfaces (Ag, Cu, Pd, Rh, and Ni), thus allowing us to create a selectivity map across various metals (Supplementary Fig. 15). Besides the elementary metal (100) surfaces, several alloys with characteristic four-fold hollow sites ($Cu_3Ag$, $Cu_3Zn$, CuAg, CuZn, and $Ni_5Ga_3$(111)), as well as various Cu facets (Cu(111), Cu(211), Cu(511), Cu(310), and Cu(110)), were investigated and shown on Fig. 5. We note that the $U$-dependent activation energies in (i), (ii), and (iii) lead to differences in the selectivity maps at varying potentials (Supplementary Fig. 16). Similar maps based on ($\Delta G_{CO^*}$, $\Delta G_{OH^*}$) have been introduced recently to describe selectivity towards various $C_1$ products[42]. Also, maps based on ($\Delta G_{CO^*}$, $\Delta G_{C^*}$) have been established to understand the $C_2/C_1$ selectivity, offering additional constraints to the optimal $\Delta G_{C^*}$ range that allows facile $C_2$ production from $C^*$[35]. Such constraints (as indicated by the vertical dashed lines) should be implemented as a prerequisite for the application of ($\Delta G_{C^*}$, $\Delta G_{OH^*}$) maps in Fig. 5. Note that Fig. 5 only addresses the $C_2$ Oxy/HC selectivity and not the activity.

The map in Fig. 5 clearly reveals that weakening the $C^*$ binding strength is the major driving force towards $C_2$ Oxy formation, offering a different interpretation than studies where O/OH binding or oxophilicity are argued to control the $C_2$ Oxy/HC selectivity[31]. For instance, despite the comparable oxophilicity, stepped $Ni_5Ga_3$(111) possesses a significantly stronger $\Delta G_{C^*}$ than Cu(211) that leads to predominant $C_2$ HC formation as experimentally probed on Ni-Ga catalysts[43]. Cu-Ag[9,18] and Cu-Zn alloys[17] were found to selectively catalyze $CO_{(2)}R$ to $C_2$ Oxy products, which also can be rationalized through the principle of lowering the $C^*$ affinity. Besides the $C^*$ affinity, the oxophilicity serves as a secondary descriptor that can help guide the selectivity of ethanol over less reduced $C_2$ Oxy products such as acetate and acetaldehyde. Enhancing the oxophilicity essentially favors further reduction of intermediates with surface-

tethered O atom, e.g. $CH_2CO^*$ and $OCHCH_3^*$ compared to desorption[44,45].

One limitation to the selectivity map in Fig. 5 is that the decision boundaries are not hard boundaries. Most of the Cu facets are located close to the boundaries between the $C_2$ HC- and the $C_2$ Oxy-selective regions, highlighting their ability to produce a variety of $C_{2+}$ products, but also the immense challenge of engineering Cu catalysts for $CO_{(2)}R$ with a single product selectivity. Pure Cu(100) facets are particularly selective towards ethylene; while undercoordinated sites on stepped Cu facets can stabilize the OCCH-$H^{TS}$ and CHC-$HO^{TS}$ relative to CHCO-$H^{TS}$ and consequently enhance the $C_2$ Oxy selectivity. Cu(110) is predicted to show the highest $C_2$ Oxy selectivity among these Cu facets, which has been experimentally validated on single-crystal Cu materials[12,13]. Cu(111) is another exception due to its extremely weak $\Delta G_{C^*}$ compared to other Cu facets, indicating the dominance of $CH_2CO$ pathway that exclusively leads to $C_2$ Oxys. Luc et al. designed Cu(111)-preferential nanosheet catalysts for selective acetate production through alkaline COR, aligning extremely well with our prediction[8]. However, the challenges for Cu(111)-type catalysts would be (1) more negative potential required to drive $C^*$ formation that also favors methane production[28,35] and (2) the strong tendency to reconstruct to Cu(100)-like facets[32]. Considering the extreme product sensitivity to the specific site motifs, investigating the stability of these motifs under operando conditions clearly needs more attention.

In conclusion, we have established the relevant reaction pathways for $CO_{(2)}R$ towards major $C_2$ Oxy/HC products of ethylene, ethanol, and acetic acid based on first principles reaction energetics and micro-kinetic modeling. We have identified a common intermediate, CHCO*, in $CO_{(2)}R$ that trifurcates to all major $C_2$ products. In contrast to previous studies, we find that the most kinetically relevant step towards ethylene formation is through the protonation of the O atom in CHCO* towards CHCOH* and subsequent dehydroxylation, which exclusively results in $C_2$ HCs. We find that protonation of the C atoms in CHCO* and the simultaneous formation of a surface–O bond leads predominantly to $C_2$ Oxy products such as ethanol or acetate. We also elucidate the puzzling pH effects in experimentally observed $C_2$ Oxy/HC selectivity. The notable pH effect arises from the additional dehydroxylation step required for complete removal of the O atom in the $C_2$ HC pathway. Alkaline conditions were found to suppress the kinetically feasible but thermodynamically less favorable pathways towards $C_2$ HCs at low overpotential. We have used scaling relations of transition state energies of key steps to determine the $C_2$ Oxy/HC selectivity with two thermodynamic descriptors: the adsorption energy of $C^*$ and $OH^*$. We found that carbophilicity is the primary feature controlling the $C_2$ Oxy/HC selectivity, while oxophilicity can be similarly important. The trends in $C_2$ Oxy/HC selectivity revealed in this work, unveils the sensitivity of $C_2$ Oxy formation on the electrolyte pH and the binding affinities of a catalytic surface and thus identifies the enormous challenges of obtaining a specific $C_2$ product through $CO_{(2)}R$. To overcome the above challenges, our descriptor-based approach presents a simple yet efficient way of deciphering the product-specific sites on Cu surfaces and designing new electrocatalysts more selective than pristine Cu.

## Methods
**Adsorption free energy calculations.** Reaction energetics were calculated with density functional theory (DFT) with a periodic plane-wave implementation and ultrasoft pseudopotentials using the QUANTUM ESPRESSO code[46], interfaced with the Atomistic Simulation Environment (ASE)[47]. We applied the BEEF-vdW functional, which provides a reasonable description of van der Waals forces while maintaining an accurate prediction of chemisorption energies[48]. Plane-wave and density cutoffs were 500 and 5000 eV, respectively, with a Fermi-level smearing

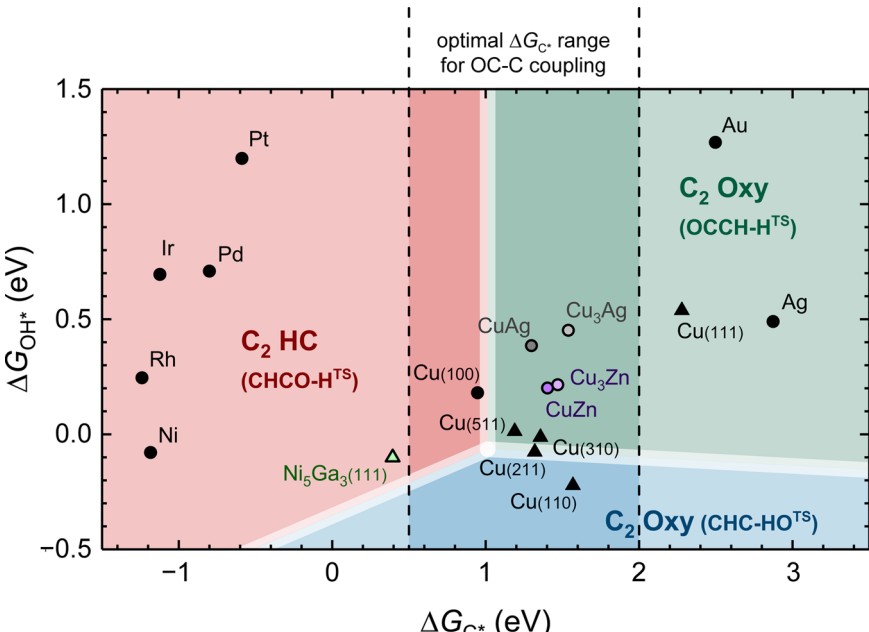

**Fig. 5 Principles of tuning the C₂ Oxy/HC selectivity through the ($\Delta G_{C^*}$, $\Delta G_{OH^*}$) map.** The ($\Delta G_{C^*}$, $\Delta G_{OH^*}$) selectivity map at $U_{RHE} = -0.73$ V at pH7. The CHCOH−, OCHCH−, and CH₂CO−dominated regions are shown as red, blue, and green, respectively, with the rate-controlling TS structures shown in the parentheses. The vertical dashed lines identify the optimal $\Delta G_{C^*}$ range for OC-C coupling according to the prior work[35]. It should also be noted that the boundaries between the three regions shift with potential due to the different $\beta$ value for the three protonation reactions. Note that the line width of these boundaries does not directly reflect the DFT errors. Source data are provided as a Source Data file.

width of 0.1 eV. psLib ultrasoft pseudopotentials were chosen. The adsorption energies on (100) surfaces of *fcc* transition metals were evaluated using four-layer (3 × 3) supercells with the bottom two layers constrained and a vacuum layer of 20 Å, and [4 × 4 × 1] Monkhorst-Pack **k**-point grids[49] were used. All the cell sizes and corresponding Monkhorst-Pack **k**-point grids for other Cu facets and intermetallic surfaces could be found in Table S1. More details about the construction of the specific metal alloy surfaces were discussed in Supplementary Note 1. All structures were optimized until the force components were less than 0.05 eV Å⁻¹. A dipole correction was applied to decouple the electrostatic interaction between the periodically repeated slabs. Gibbs free energy correction was applied using the harmonic oscillator approximation. Solvation correction was applied by comparing the adsorption energy calculated in a three-layer (4 × 3) supercell with or without a monolayer of explicit water on a Cu(100) slab. Detailed methods and parameters for the Gibbs free energy correction and solvation correction were shown in Supplementary Note 2 and Supplementary Note 3. Solvation corrections were calculated explicitly for intermediates where there are −O or −OH groups that can face towards the solvent. For small C₁ intermediates and C₂ intermediates that lie close to the surface, we assume negligible (−0.1 eV) or no (0 eV) interaction with the intermediates (as explained in Supplementary Note 3). Although strong solvation effects on OH* adsorption have been widely discussed in literature[50,51], it is not currently possible to judge if the value chosen for $\Delta G_{OH^*}$ in this work is corrected or not. However, significant deviations will not affect the conclusion of this work because $\Delta G_{OH^*}$ only serves as a descriptor; OH solvation corrections would merely lead to a constant shift for the y-axis in the selectivity map of Fig. 5 while not changing the relative positions of and the trends among different materials. In addition, exchange-and-correlation functional that accounts for long-range interactions, such as BEEF-vdW, generally results in a decrease in the strength of solvation contributions to the adsorption energies, providing a possible explanation to the relatively small solvation corrections we used in this work[52,53]. To include the predominant effect of interfacial electric field on stabilizing some C₂ intermediates, an electric field model was constructed following the same method described in ref. [35]. The detailed parameters were shown in Supplementary Note 4.

**Electrochemical reaction barrier calculations**. Electrochemical barriers were calculated with (4 × 3) supercells and Monkhorst-Pack k-point grids of [3 × 4 × 1], respectively. All structures contained a three-layer transition metal slab, with atoms in the top layer relaxed and the rest fixed, along with a hydrogen-bonded water layer determined through minima hopping[54]. Further details regarding minima hopping are provided in the Supplementary Note 5. Transition state geometries and energies were calculated using the climbing-image nudged elastic band (CI-NEB) method, with the forces on the climbing image converged to less than 0.05 eV Å⁻¹ (ref. [55]).The spring constants were tightened for images close to the

saddle point[56]. The plane wave and charge density cutoff, exchange-correlation functional, and other parameters were the same as those used for geometry optimizations. The charge extrapolation method[57] was used to deduce the activation barriers at constant potential. The Bader charge and work function for each state were calculated to estimate the energy change induced by charging. For simplicity and consistency across the literature, Bader analysis was applied as the primary spatial partitioning scheme in this study[25,57]. Voronoi and Hirschfeld analyses should be technically compatible with the charge extrapolation scheme, which, however, are suggested to be considered in future works. A hydronium ion was present in the initial state to act as the hydrogen source for protonation. After relaxing NEB calculations, Bader charges of the initial state, transition state, and final state are calculated by partitioning the proton-adsorbate complex such that the proton remains as part of the solvent, and using a z-plane threshold that separates the immediate water layer from the metal slab within the unit cell. The net charge of the initial state, transition state, or final state can then be determined via: *net_charge = (slab_charge + ads_charge) − (neutral_slab_charge + neutral_ads_charge)*. Where *neutral_slab_charge* and *neutral_ads_charge* are determined as a sum of the valence number for a particle molecule/metallic slab. The effect of utilizing hydronium and water as proton sources, respectively, was discussed in detail in Supplementary Note 5. Since challenges remain for modeling electrochemical activation barriers, the underlying limitations/assumptions from our methodology are listed in Supplementary Note 9.

**Microkinetic modeling**. Mean-field microkinetic models were simulated with the CatMAP software package[58]. The CatMAP software package used in this work can be accessed and downloaded through https://github.com/SUNCAT-Center/catmap. All the elementary steps were described in Supplementary Note 6. Adsorbate-adsorbate interactions were considered for all possible reaction intermediate pairs, using the methods in Liu et al.[25] in which first order adsorbate-adsorbate interactions cause a linear weakening in adsorption energies of a given intermediate when the coverage of adsorbates exceed a certain threshold (0.25 monolayers in this work). Based on previous works[25,32], CO is the predominant intermediate on the Cu surface. We assume that there will be minimal coverage effects on reaction barriers and energetics aside from interactions with CO, which have been included in the model. Further details regarding micro kinetic modeling with adsorbate-adsorbate interactions are explained in Supplementary Note 6.

## Data availability

The atomic structures data and corresponding energetics generated in this study have been deposited in the publically available Catalysis-hub[59] repository database at https://www.catalysis-hub.org/publications/PengTrends2022. Source data are provided with this paper.

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

## Acknowledgements

The material for all the energetic computation and the microkinetic models on Cu is based upon work performed by the Joint Center for Artificial Photosynthesis, a DOE Energy Innovation Hub, supported through the Office of Science of the U.S. Department of Energy under Award Number DE-SC0004993. The material for energetic computation on materials other than Cu is based on work performed by the Liquid Sunlight Alliance, which is supported by the U.S. Department of Energy, Office of Science, Office of Basic Energy Sciences, Fuels from Sunlight Hub under Award Number DE-SC0021266. We acknowledge the use of the computer time allocation for the Material Simulations in Joint Center for Artificial Photosynthesis (JCAP) at the National Energy Research Scientific Computing Center, a DOE Office of Science User Facility supported by the Office of Science of the U.S. Department of Energy under Contract No. DE-AC02-05CH11231. J.H.S acknowledge funding from the Knut and Alice Wallenberg Foundation (grant nr. 2019.0586). The authors thank Dr. Tao Wang and Dr. Lei Wang for insightful discussions and helpful suggestions.

## Author contributions

H.-J.P., M.T.T., and F.A.-P. conceived the research idea and designed the computational models. H.-J.P. performed all the DFT calculations and analyzed all the data. H.-J.P and X.L. performed microkinetic modeling. H.-J.P., M.T.T., and X.L. analyzed all the data. J.H.S. and F.A.-P. discussed the results and commented on the manuscript. H.-J.P. wrote and revised the paper. M.T.T., J.H.S., and F.A.-P. took part in revising the paper.

## Competing interests

The authors declare no competing interests.
