## [Peer Review File · Nature Communications]

Reviewer #1 (Remarks to the Author):

The paper by Frank Abild-Pedersen and coworkers on C2 selectivity trends toward hydrocarbons vs oxygenates using DFT calculations and microkinetic modelling will likely be publishable in Nature Communications upon major changes.

I have two main concerns and some requests.

Concern 1: BEEF vdW was used, which contains large gas-phase errors. Reading from a recent work by the Koper group (ACS Catal. 2020, 10, 12, 6900–6907) and Vegge's group (Catal. Sci. Technol., 2015, 5, 4946–4949), there are large errors using BEEF vdW for CO, hydrocarbons, carbonyl compounds, carboxylic acids, etc. Thus, the overall reaction energies (from CO to the final products) and the associated equilibrium potentials considered by Abild-Pedersen et al are either far from experiments or close to them because of error cancellation. In Table S2 I see that only H₂, acetic acid and acetaldehyde were corrected. Hence, the authors should calculate their equilibrium potentials, check that they are fine for the right reason or otherwise make the necessary corrections.

Concern 2: Some of the solvation corrections in Table S4 are too small. For instance, the values for *COH (-0.11 eV) and *OH (0 eV) differ drastically with respect to many others in the literature, either implicit, explicit, or micro-solvated, using static or dynamic methods (Energy Environ. Sci., 2010, 3, 1311–1315; Appl. Catal. B 276, 2020, 119147; J. Chem. Inf. Model. 2019, 59, 5, 2190–2198; Catal. Today 323, 2019, 35–43). If C1 adsorbates are not well described, I wonder if C2 adsorbates are fine and if it all depends on the configuration of the water layer with respect to the adsorbates. Hence, the authors should calculate the solvation corrections using another method to ensure that solvation is well taken care of.

Requests:

1. Please explain how the minima hopping is used here and if the results depend strongly on the configurations found with it. It is a constant that groups using the charge extrapolation method do not really make the reader aware about the importance of this. In addition, are there any problems with charge localization? Must the Bader method be used or would other methods (Voronoi, Mulliken, Hirshfeld) render similar results?
2. Explain in detail what DSC is and provide the equations necessary to use it.
3. How do the authors calculate the beta factors from Bader charges? Do they subtract the charge in the initial state? Must the Bader method be used or would other methods (Voronoi, Mulliken, Hirshfeld) render similar results? How do the authors' results compare to other methods (J. Phys. Chem. Lett. 2021, 12, 21, 5193–5200; Angew. Chem. Intl. Ed. 52, 2013, 2459–2462)?
4. Figures 3–4: Very important, please use a conventional x-axis (more negative numbers to the left). Currently, the figures are currently rather confusing.

Reviewer #2 (Remarks to the Author):

Density functional calculations are used in this study to determine reaction energies and activation barriers of elementary steps involved in CO/CO₂ reduction on Cu(100) surfaces. Emphasis is placed on steps involved in dictating the selectivity of which C₂ products are formed. Microkinetic modeling is used to help clarify dominant pathways and what steps dictate selectivity. Overall, this is a competent, detail oriented study of reaction paths, and certainly worthwhile to publish. However, there are

significant limitations that make this a relatively incremental contribution. I am not sure the standards of Nature Communications, but the limitations mentioned below and overall lack of anything novel makes this a paper suitable for an archival catalysis, electrochemistry, or physical chemistry journal. The two main limitations of this work that limit its impact:

1) The methods used, as detailed in the SI, are neither novel nor reliable to provide reaction energetics accurate enough to make definitive conclusions as to reaction mechanisms. All methods used are previously published, so there are no method advances in this work. The DFT calculations use a single adsorbate coverage for all adsorbates, though it is unknown to what extent coverage dependences will impact reaction energetics for all the steps involved, or whether the low coverage used represents the operating surface at all electrode potentials considered. Solvation and electrification effects are difficult to represent in electrochemical systems, and can have large effects on reaction energetics that would alter conclusions. Solvation has been included in an entirely heuristic way with constant values roughly assumed for different adsorbates based on their structures. Electrification assumes an arbitrary Helmholtz model, does not consider PZC effects of reaction conditions, and makes an arbitrary (and quantitatively important) choice for the distance of charge separation. The approach to calculate barriers suffers from similar assumptions/approximations, and does not successfully establish whether the initial state to which barriers are referenced properly captures the chemical potential of the bulk phase ion reactant. The assumptions made as to how energetics are altered whether the proton donor is a hydronium ion or water molecule essentially “bake in” the conclusions as to pH effects rather than allowed the model to analyze these effects. Though all of these limitations are inherent to the established approaches used in this study to model electrocatalytic energetics, when even further coupled with inherent DFT error (and lack of consideration of interfacial entropic effects), the methods used are collectively not precise enough to answer the research questions posed. The methods used have been competently executed, but simply suffer too substantially from the inherent difficulties in DFT representation of electrocatalytic energetics to reach definitive and substantive conclusions. Further, much of these approximations are only apparent to the expert (and mainly noted on a detailed read of the SI), and the manuscript itself does not provide a sufficient quantification or critical discussion of these limitations.

2) The electroreduction of CO or CO₂ on Cu electrodes, empirically, does not provide sufficient selectivity to any C₂ product to be of any practical use. The work done here is only directly relevant to a single electrode surface, for which numerous DFT studies already exist, and makes the relatively minor advance of considering more details of specific C₂ product-determining steps. Given the lack of practical relevance of the electrode studied, and incremental nature of the mechanistic advances, the impact of further detailed DFT work on CO/CO₂ reduction on Cu is minimal. Frankly, there seems to be a self-perpetuating cycle of publications in this area without any significant practical advance beyond well established knowledge on the performance of Cu electrodes for CO₂ electroreduction.

Reviewer #3 (Remarks to the Author):

Electrocatalytic CO and CO₂ reduction is a hot topic; there are every month new papers in high impact journals on this. The interest is twofold: firstly, it would be a breakthrough in power-to-x technology if these reactions are catalyzed in a selective and efficient manner, secondly, the chemistry on a Cu

surface is very rich and complex, which means that it is a challenge to obtain fundamental insight. So far, nobody has found a great novel catalyst for this reaction. Therefore, most papers, including this one, address the understanding of the reactions. The carbon-carbon bond that can form on Cu is a puzzle and understanding the selectivity is from a fundamental standpoint very interesting.

The amount of work behind this paper is huge and the analysis is careful. In the analysis of electrochemical reactions there are a number of choices done to setup the model and reaction path. However, the results are very sensitive to the exact manner in which the model is constructed. This means that it is difficult to obtain conclusive insights.

This is reflected in Fig 5. Cu 211, 511, 310 are very different in measured selectivity, however, in OH and C binding they are very close to each other. This probably means that there is something, not considered in the model, responsible for the differences in experiments for those facets. I would like the author to comment on that.

I also think the authors could be more clear in their claims, however, the paper has a high quality due to the amount of data and the careful treatment. I can therefore recommend that the paper is accepted in Nature Com.

Reviewer #1 (Remarks to the Author):

The paper by Frank Abild-Pedersen and coworkers on C2 selectivity trends toward hydrocarbons vs oxygenates using DFT calculations and microkinetic modelling will likely be publishable in Nature Communications upon major changes.

I have two main concerns and some requests.

Concern 1: BEEF vdW was used, which contains large gas-phase errors. Reading from a recent work by the Koper group (ACS Catal. 2020, 10, 12, 6900–6907) and Vegge's group (Catal. Sci. Technol., 2015, 5, 4946-4949), there are large errors using BEEF vdW for CO, hydrocarbons, carbonyl compounds, carboxylic acids, etc. Thus, the overall reaction energies (from CO to the final products) and the associated equilibrium potentials considered by Abild-Pedersen et al are either far from experiments or close to them because of error cancellation. In Table S2 I see that only H₂, acetic acid and acetaldehyde were corrected. Hence, the authors should calculate their equilibrium potentials, check that they are fine for the right reason or otherwise make the necessary corrections.

We thank the reviewer for the concern regarding BEEF-vdW corrections with respect to the gas-phase energies. The original paper from Christensen et al. (Catal. Sci. Technol. 2015, 5, 4946) provided a correction scheme that alleviates the functional's systemic error, which we have now employed throughout the study. As the reviewer suggested, we have also calculated the equilibrium potentials and compare with what summarized in a recent Review paper (Nitopi et al. Chem. Rev. 2019, 119, 7610) to assure readers that using BEEF-vdW with corrections provide energetics with reasonable accuracies. Note that the gas-phase energies present in the original Table S2 are not all obtained at standard conditions (298 K, 101325 Pa), which will result in deviations from experimental equilibrium potentials, as the reviewer pointed out. In the revised Table S2, we have supplemented standard free energies of all gas-phase and dissolved species for more straightforward comparison. All the details about the corrections to gas-phase molecules have been shown in the revised **Supplementary Note 2** and appended **Table S3** (Page S2–S4, SI). The errors in standard reaction enthalpy and equilibrium potential are within ± 5 kJ mol⁻¹ and ± 0.02 V. The small errors indicate that our correction scheme sufficiently describe the reaction thermodynamics with acceptable accuracy. The revised corrections to carbonyl compounds such as CH₂CO, CH₃CHO, and CH₃COOH have slightly changed the free energy diagrams involving these species (**Figures 2d, 2e, and S10**). Nevertheless, the new correction scheme does not influence the kinetic modeling results and the main conclusions in our paper.

In addition to the revised figures and tables, the following changes were made to the **Supplementary Note 2** (changes in blue font):

“Here we made the same assumption as Peterson *et al.* did in 2010⁷ that gaseous products in the pathway were calculated at partial pressures corresponding to the Faradaic yields reported by Hori et al,⁸ and liquid (or dissolved) products in the

pathway were calculated at a molarity of 1 mol L⁻¹ (M). The fugacities we used for gaseous products are the same as in Ref.⁷. Acetylene (C₂H₂), ethane (C₂H₆), ethenone (CH₂CO), acetic acid (CH₃COOH), acetaldehyde (CH₃CHO), and vinyl alcohol (CH₂CHOH) are not counted for Faradaic yields in Ref.⁸. We then assumed that all stable C₂ HC (i.e. C₂H₂ and C₂H₆) have the same fugacity as ethylene (C₂H₄), all stable C₂ Oxy except ethanol (CH₃CH₂OH), i.e. CH₃COOH and CH₃CHO, have the fugacities calculated based on Henry's law, and the two unstable molecules of CH₂CO and CH₂CHOH have a very low fugacity of 0.1 Pa. The constants applied in Henry's law, for CH₃COOH and CH₃CHO, are 1.82×10⁻⁴ and 6.67×10⁻² bar M⁻¹,⁹ corresponding to fugacities of 18.2 and 6670 Pa for CH₃COOH and CH₃CHO, respectively. Therefore, the contribution to the chemical potential μ of gas-phase or dissolved molecules could be found in **Table S2**. Note that a DFT correction of +0.15 eV was added for gas-phase molecules containing C=O double bonds (CH₂CO, CH₃COOH, and CH₃CHO), according to Christensen *et al.*¹⁰, and a DFT correction of +0.09 eV was added for hydrogen gas (H₂) due to the systematic error caused by the BEEF-vdW functional, in accordance with Studt *et al.*¹¹” (Page S2, SI)

“To validate the afore obtained gas-phase energies, the energies at the standard condition (298 K, 101325 Pa) are used to calculate the equilibrium potentials for comparison with the experimental values obtained from the NIST Chemistry Webbook (<https://webbook.nist.gov/chemistry/>) and Langes Handbook of Chemistry. Most of the experimental data has been collected in the Supporting Information of Ref.⁹ The standard reaction enthalpies (ΔH^0) and equilibrium potentials (U_{eq}) are compared in **Table S3**. It is notable that the errors in ΔH^0 and U_{eq} are within ± 5 kJ mol⁻¹ and ± 0.02 V, respectively. Ethenone (CH₂CO) is an exception herein and the errors in ΔH^0 are calculated as 18.65 and -20.55 kJ mol⁻¹, depending on the experimental data from different sources. The experimentally measured ΔH^0 varies by ~ 40 kJ mol⁻¹. Overall, these results indicate that with the above DFT corrections applied to H₂, CH₂CO, CH₃COOH, and CH₃CHO, reasonable agreement with thermochemical data is achieved.” (Page S3, SI)

Concern 2: Some of the solvation corrections in Table S4 are too small. For instance, the values for *COH (-0.11 eV) and *OH (0 eV) differ drastically with respect to many others in the literature, either implicit, explicit, or micro-solvated, using static or dynamic methods (Energy Environ. Sci., 2010,3, 1311-1315; Appl. Catal. B 276, 2020, 119147; J. Chem. Inf. Model. 2019, 59, 5, 2190–2198; Catal. Today 323, 2019, 35-43). If C1 adsorbates are not well described, I wonder if C2 adsorbates are fine and if it all depends on the configuration of the water layer with respect to the adsorbates. Hence, the authors should calculate the solvation corrections using another method to ensure that solvation is well taken care of.

We thank the reviewer for the concern. Solvation effects are still a relatively new and controversial topic within electrochemical reduction, and thus correction schemes vary wildly in the literature. In addition, previous benchmark works have revealed

that exchange-and-correlation functional that accounts for long-range interactions, such as BEEF-vdW, generally results in a decrease in the strength of solvation contributions to the adsorption energies (Wellendorff et al. J. Chem. Phys. 2014, 140, 144107; Granda-Marulanda et al. ChemPhysChem 2019, 20, 2968). This provides a possible explanation to the relatively small solvation corrections we used in this work. The solvation corrections we use follow closely with our past studies (Appl. Catal. B 2020, 279, 119384; Energy Environ. Sci. 2021, 14, 473) and aligns well with works by researchers who have used similar methods (Calle-Vallejo et al. Angew. Chem. Int. Ed. 2013, 52, 7282; Montoya et al. J. Phys. Chem. Lett. 2015, 6, 2032; Liu et al. Nat. Commun. 2019, 10, 32; Ludwig et al. J. Phys. Chem. C 2019, 123, 5999; etc.). The COH* and OH* corrections can be considered small but they are not the main focus of the paper; changing these values do not significantly impact the conclusions regarding C₂ oxygenates and hydrocarbons as discussed in the paper. We clarify to the reader how we include solvation corrections and discuss the limitations of our approach in the revised manuscript.

The following changes were made to the manuscript method (changes in blue font):

“Solvation corrections were calculated explicitly for intermediates where there are –O or –OH groups that can face towards the solvent. For small C₁ intermediates and C₂ intermediates that lie close to the surface, we assume negligible (–0.1 eV) or no (0 eV) interaction with the intermediates (as explained in **Supplementary Note 3**).” (Page 19, Main Text)

“Although strong solvation effects on OH* adsorption have been widely discussed in literature,^{51,52} it is not currently possible to judge if the value chosen for ΔG_{OH^*} in this work is corrected or not. However, significant deviations will not affect the conclusion of this work because ΔG_{OH^*} only serves as a descriptor; OH solvation corrections would merely lead to a constant shift for the y-axis in the selectivity map of **Figure 5** while not changing the relative positions of and the trends among different materials. In addition, exchange-and-correlation functional that accounts for long-range interactions, such as BEEF-vdW, generally results in a decrease in the strength of solvation contributions to the adsorption energies, providing a possible explanation to the relatively small solvation corrections we used in this work.^{53,54}” (Pages 19–20, Main Text)

Since this work primarily focuses on intermediates stemming from the reduction of CCO*. We have explicitly analyzed the solvation effects on adsorption energies of key intermediates of CHCO*, CH₂CO*, OCHCH* and CHCOH* using the model containing a metal slab and a layer of explicit water (please see **Supplementary Note 3** for details). The adsorption energies of these species at vacuum and solvation conditions are presented in the following table.

Table R1. Adsorption electronic energy in vacuum and solvation conditions

Specie	Adsorption energy (eV, vacuum)	Adsorption energy (eV, with explicit solvents)	Difference (eV)

CHCO*	-1.76	-1.87	-0.11
CH ₂ CO*	-2.00	-2.07	-0.07
OCHCH*	-2.39	-2.45	-0.06
CHCOH*	-1.68	-1.78	-0.10
CCH*	-1.41	-1.46	-0.05
CHCHOH*	-2.01	-2.12	-0.11

For simplicity, we used a constant -0.10 eV solvation correction for these species, and variations within a few tenths of meV (as shown in **Table R1**) will not impact the conclusions of this work. While for other species that we simply assume to have a similar correction, the minimum change in solvation correction required to alter our conclusion will be $(0.78 \text{ eV} - 0.40 \text{ eV}) = 0.38 \text{ eV}$ if we read the smallest difference in barrier heights from **Figure S6**. This corresponds to the selectivity determining protonation of the oxygen, or the primary (or secondary) carbon of the CHCOH* intermediate. However, we performed additional calculations using the explicit solvation method and could find in **Table R1** that the difference in solvation corrections between the two relevant species CCH* and CHCHOH* is still within a few tenths of meV, further supporting our assumption and conclusions.

We have added the following statement to the manuscript (**changes in blue font**):

“Three-layer (4×3) supercells with a monolayer of explicit water, *i.e.* the same setting for electrochemical barrier calculations, were employed to investigate the solvation effect on several key intermediates identified in this work, including CH₂CO*, OCHCH*, CHCOH*, **CHCHOH***, **CCH***, CH₂CHOH*, and OCHCH₃*. Except CH₂CHOH* and OCHCH₃*, the studied **five** important C₂ intermediates have a similar $E_{\text{solv corr}}$ of around -0.10 eV, while CH₂CHOH* and OCHCH₃* have a distinct $E_{\text{solv corr}}$ of -0.51 eV and -0.25 eV, respectively. Due to the similar binding geometry, OCH₂CH₃* was assumed to have the same $E_{\text{solv corr}}$ of -0.25 eV as OCHCH₃*. The rest of C₂ species were assumed to have a $E_{\text{solv corr}}$ of -0.10 eV according to the results obtained for CH₂CO*, OCHCH*, CHCOH*, **CHCHOH***, **and CCH***.” (Page S5, SI)

Requests:

1. Please explain how the minima hopping is used here and if the results depend strongly on the configurations found with it. It is a constant that groups using the charge extrapolation method do not really make the reader aware about the importance of this. In addition, are there any problems with charge localization? Must the Bader method be used or would other methods (Voronoi, Mulliken, Hirshfeld) render similar results?

We thank the reviewer for the request. Minima hopping was used to determine the water structures surrounding the metal surfaces. A recent study (Chen et al. Nat. Commun. 2018, 9, 3202) determined that the partial charge of the solvated proton and provides the rationale for the use of Bader analysis despite its simplicity.

“The reason that we have used this simplified partitioning scheme is that in a capacitor model developed for electrochemical reactions, the potential dependence is only a function of the distance of the charged species from the surface (the z direction). Moreover, the excess charge is only slightly delocalized to the other water molecules within the same layer ($0.11 e$) with the majority of the positive charge residing on the hydronium ion itself (see Supplementary Fig. 4), and collapsing the x and y directions affords a clearer picture of the charge separation at the electrochemical double layer.” (Page 3, Chen et al. Nat. Commun. 2018, 9, 3202)

Technically, other methods should work as well (Voronoi and Hirschfeld both also use spatial partitioning of the electronic density, just differently). If we wanted to benchmark these methods, we feel this would be a paper in its own right.

We have added the following changes (changes in blue font):

“For simplicity and consistency across the literature, Bader analysis was applied as the primary spatial partitioning scheme in this study.^{25, 58} Voronoi and Hirschfeld analyses should be technically compatible with the charge extrapolation scheme, which, however, are suggested to be considered in future works.” (Page 20, Main Text)

2. Explain in detail what DSC is and provide the equations necessary to use it.

We thank the reviewer for the request. The DSC is explained in detail in the revised SI. The **equations (18–20)** are also provided in SI. Please find these details in **Supplementary Note 7** as (changes in blue font):

“The DSC is a concept derived from the degree of rate control (DRC), which was conceptualized by Campbell *et al.*^{42, 43} The DRC is a mathematical approach for analyzing reaction mechanisms and kinetics of multistep reactions. The DRC for elementary step i , $X_{RC,i}$, is initially defined as

$$X_{RC,i} = \frac{k_i}{r} \left(\frac{\partial r}{\partial k_i} \right)_{k_{j \neq i}, K_i} = \left(\frac{\partial \ln r}{\partial \ln k_i} \right)_{k_{j \neq i}, K_i} \quad (18)$$

where r is the net reaction rate to the product of interest, and the partial derivative is taken holding constant the rate constants, k_j , for all other steps $j \neq i$ and the equilibrium constant, K_i , for step i (and all other steps too, since their forward and reverse rate constants are held fixed). By this definition above, $X_{RC,i}$ equals the relative increase in the net rate per relative increase in the rate constant for step i (differentially). The larger the numeric value of $X_{RC,i}$ is for a given step, the bigger is the influence of its rate constant on the overall reaction rate r . A positive value indicates that increasing k_i will increase the net rate r ; such steps are termed rate-limiting steps. A negative value indicates the opposite; such steps are termed inhibition steps. While Eq. (18) looks like equations used in standard differential sensitivity analyses, it ensures thermodynamic and kinetic consistency as the

entities in this partial derivative are held constant. By further correlating k_i to reaction thermodynamics and kinetics using transition state theory, $X_{RC,i}$ could be expressed in a more general way, the generalized degree of rate control of species i , DRC_i , as:

$$DRC_i = \frac{1}{r} \left(\frac{\partial r}{\partial \left(\frac{-G_i^0}{RT} \right)} \right)_{G_{j \neq i}^0} = \left(\frac{\partial \ln r}{\partial \left(\frac{-G_i^0}{RT} \right)} \right)_{G_{j \neq i}^0} = \left(\frac{-\partial \ln r}{\partial \left(\frac{G_i^0}{RT} \right)} \right)_{G_{j \neq i}^0} \quad (19)$$

where the partial derivative is now taken holding constant the standard-state free energy of all other species (intermediates, transition states, reactants, and products), j . Its value describes the relative increase in net rate due to the (differential) stabilization of the standard-state free energy for species i , holding all the other species' energies constant. Thus, Eq. (19) probes the importance of one species' free energy in the full standard-state free-energy surface for the full reaction.

Similarly, the generalized DSC of species i , DSC_i , could be defined as the sensitivity of the selectivity, $S = r_P/r_R$, where r_P is the rate of production of the desired product P and r_R is the rate of consumption of a reactant R, to energies of i as:

$$\begin{aligned} DSC_i &= \frac{1}{S} \left(\frac{\partial S}{\partial \left(\frac{-G_i^0}{RT} \right)} \right)_{G_{j \neq i}^0} = \left(\frac{\partial \ln S}{\partial \left(\frac{-G_i^0}{RT} \right)} \right)_{G_{j \neq i}^0} = \left(\frac{-\partial \ln S}{\partial \left(\frac{G_i^0}{RT} \right)} \right)_{G_{j \neq i}^0} \\ &= \left(\frac{-\partial \ln(r_P/r_R)}{\partial \left(\frac{G_i^0}{RT} \right)} \right)_{G_{j \neq i}^0} = \left(\frac{-\partial \ln(r_P)}{\partial \left(\frac{G_i^0}{RT} \right)} \right)_{G_{j \neq i}^0} - \left(\frac{-\partial \ln(r_R)}{\partial \left(\frac{G_i^0}{RT} \right)} \right)_{G_{j \neq i}^0} \\ &= DRC_{i,P} - DRC_{i,R} \end{aligned} \quad (20)$$

where $DRC_{i,P}$ and $DRC_{i,R}$ are the degrees of rate control of species i for the rates of making P and consuming R, respectively. The value of DSC_i describes the relative increase in net selectivity to P from R due to the (differential) stabilization of the standard-state free energy for species i holding all other species' energies constant. The DSCs shown in **Figure 4** are all calculated using $(DRC_{i,P} - DRC_{i,CO})$ where P is ethylene, ethanol, or acetic acid. The $DRC_{i,P}$ and $DRC_{i,CO}$ were mathematically obtained using the code implemented in the CATMAP software package.³⁶ (Pages S17–S18, SI)

3. How do the authors calculate the beta factors from Bader charges? Do they subtract the charge in the initial state? Must the Bader method be used or would other methods (Voronoi, Mulliken, Hirshfeld) render similar results? How do the authors' results compare to other methods (J. Phys. Chem. Lett. 2021, 12, 21, 5193–5200;

Angew. Chem. Intl. Ed. 52, 2013, 2459-2462)?

We thank the reviewer for the request. Regarding charge partitioning, this question relates to the first request and we reiterate that, in principle, either of the suggested schemes could be used. For consistency with the previous literature, we use Bader charges. Beta factors are calculated based on how much charge has been transferred from the transition state relative to the initial state. In our study, we observe a partial charge of around 0.7 in agreement with the following study (Chen et al. Nat. Commun. 2018, 9, 3202). During the transition state, some charge is transferred over to the adsorbate/metal surface, which leaves a different charge on the solvated proton as it is forming bonds.

We have added the following changes (changes in blue font):

“After relaxing NEB calculations, Bader charges of the initial state, transition state, and final state are calculated by partitioning the proton-adsorbate complex such that the proton remains as part of the solvent, and using a z-plane threshold that separates the immediate water layer from the metal slab within the unit cell. The net charge of the initial state, transition state, or final state can then be determined via: $net_charge = (slab_charge + ads_charge) - (neutral_slab_charge + neutral_ads_charge)$. Where $neutral_slab_charge$ and $neutral_ads_charge$ are determined as a sum of the valence number for a particle molecule/metallic slab.” (Page 20, Main Text)

One of the mentioned paper is from Chan and Nørskov’ group (Patel et al. J. Phys. Chem. Lett. 2021, 12, 5193). In this original paper, the proposed “force” method results in similar barrier heights for electrochemical protonation reactions to the charge-extrapolation “Bader” method (please see Figures 2 and 3 in Patel et al. J. Phys. Chem. Lett. 2021, 12, 5193). It is definitely valuable to evaluate the protonation barriers of the selectivity determining steps identified in our work using this new method. But we feel this would be a paper of its own.

The other mentioned paper is from Asthagiri and Janik’s group (Nie et al. Angew. Chem. Intl. Ed. 2013, 52, 2459). The authors proposed a H-shuttling mechanism for obtaining electrochemical barriers. We have not considered this method in this work regarding C₂ oxygenates and hydrocarbons because we have compared our protonation barriers for C₁ species to those of the Asthagiri and Janik group in our previous studies (Appl. Catal. B 2020, 279, 119384; Energy Environ. Sci. 2021, 14, 473) and found that these barriers align well. Given that the H-shuttling mechanism only considers one water molecule during the barrier estimation, we used the charge-extrapolation method that considers more water molecules for the solvation environment.

We have added the following changes to explain why we primarily considered the charge-extrapolation method for barrier estimation (changes in blue font):

“All other protonation barriers are the same as reported in our previous work.⁵ In

Ref.⁵, some reported barriers for the formation of C₁ species (e.g. CO-H protonation to form COH*) have been compared with barriers reported in previous works using different methods.^{34, 35} These barriers were found to align well; therefore, in the present work regarding C₂ Oxy/HC formation, we only considered the charge-extrapolation method, which is aiming to be consistent with previous works.^{5, 12, 28}” (Page S13, SI)

4. Figures 3-4: Very important, please use a conventional x-axis (more negative numbers to the left). Currently, the figures are currently rather confusing.

We thank the reviewer for the request. All the figures containing x-axes of potential have been revised as requested. These figures include **Figures 3, 4, S4–S6, S8–S10, S11, S12, and S14**. Figure S3 was not revised because it was directly adopted from literature. Some words related to the discussion based on these figures were changed correspondingly (changes in blue font):

“On the **right** leg ($U_{\text{RHE}} > -0.5$ V), the SDS for C₂ Oxys (mainly OCCH-H^{TS} on Cu(100)) competes with CHCOH-H^{TS}; while on the **left** leg ($U_{\text{RHE}} < -0.5$ V), the SDS for C₂ Oxys competes with CHCO-H^{TS}.” (Page 13, Main Text)

“On the **right** leg we have ...”, “whereas on the **left** leg we find ...”, “The potential and pH dependence of key intermediate energies present in Eq. (1) and Eq. (2) induces the upward shift of the inverse volcano with increasing pH on a U_{RHE} scale or, *videlicet*, the shift of the **right** leg of the volcano towards more negative potentials on a U_{SHE} scale (**Figure 4d, e**).” (Page 14, Main Text)

Reviewer #2 (Remarks to the Author):

Density functional calculations are used in this study to determine reaction energies and activation barriers of elementary steps involved in CO/CO₂ reduction on Cu(100) surfaces. Emphasis is placed on steps involved in dictating the selectivity of which C₂ products are formed. Microkinetic modeling is used to help clarify dominant pathways and what steps dictate selectivity. Overall, this is a competent, detail oriented study of reaction paths, and certainly worthwhile to publish. However, there are significant limitations that make this a relatively incremental contribution. I am not sure the standards of Nature Communications, but the limitations mentioned below and overall lack of anything novel makes this a paper suitable for an archival catalysis, electrochemistry, or physical chemistry journal.

We thank the reviewer for the careful evaluation, although we will counter the verdict. We believe the importance of our study comes from the conclusions rather than the methods used, and primarily because we focus on elucidating mechanisms on Cu. Our work attempts to address the questions of how, on the atomic scale, Cu produces ethylene and ethanol/acetate, whether these products share the same mechanistic pathways and can be represented by the same surface descriptors, and what governs selectivity between the hydrocarbon versus oxygenate products. These are essential questions to shed light on in the strive towards sustainable chemical and energy industries. This as Cu is currently the only material known to produce these products from CO₂ electrochemically in any significant amount. The closest work from Calle-Vallejo et al. (Angew. Chem. Int. Ed. 2013, 52, 7282) merely touches on the thermodynamics of the question. The kinetic insights from our work shows that it cannot be possible for ethylene to form through the CH-CHO* pathway, as the barrier to break the last -O bond is too high. While the work from Goddard's group (Cheng et al. PNAS 2017, 114, 1795) fully considered the solvation and surface-charge effects on the relevant protonation barriers using a constant-potential method, the authors mainly focused on forward barriers and overlooked the possibility of deprotonation of unstable intermediates (e.g. CCOH* to CCO*) along their reaction pathways. Moreover, according to their calculations, the barrier towards ethanol is 0.43 eV higher than ethylene, which will result in over 7 orders of magnitude difference in their formation rates at the relevant potential of -0.59 V vs. RHE (pH7). This is in disagreement with experiments. This sort of observation challenges the current mechanistic understandings of forming ethylene and ethanol. Our work fully integrates both reaction thermodynamics/kinetics at a wide potential range and enables mechanistic interpretation of pH effects on tuning oxygenate/hydrocarbon selectivity. In previous studies, more attention was paid to pH effects on C₁/C₂ selectivity rather than C₂ oxygenate/hydrocarbon selectivity, and the lack of mechanistic understandings for that limits the progress towards designing catalysts for production of liquid fuels (*cf.* oxygenates) in CO₂R. In addition, we form a novel yet simple descriptor model with C* binding energy and OH* binding energy to show that different surfaces produce different C₂ species depending on variations in these

properties. If Au/Ag were to form C₂ products, they would mainly form C₂ oxygenates based on our descriptor model. This is also in agreement with the experimentally observed enhancements in C₂ oxygenate production using Cu-Ag catalysts. An incremental contribution would usually be additional data that supports already-published descriptor models. Despite certain shortcomings of DFT, there is still a lot we can learn though using it to model electrochemical systems. Rather than producing novel methods, the quality of this work is substantiated by how detail-oriented it is. Nevertheless, as noted by the reviewer, we will add further statements on the limitations of our methods and future directions to explore. Please check the point-to-point response in details as followings.

The two main limitations of this work that limit its impact:

1) The methods used, as detailed in the SI, are neither novel nor reliable to provide reaction energetics accurate enough to make definitive conclusions as to reaction mechanisms. All methods used are previously published, so there are no method advances in this work.

We thank the reviewer for the analysis. Again, we believe the importance of our study comes from the conclusions rather than the methods used. The claim that the methods are not reliable to make definitive conclusions on reaction energetics is troubling. Our thermodynamic energetics fall closely with previously reported DFT works (Calle-Vallejo et al. *Angew. Chem. Int. Ed.* 2013, 52, 7282; Cheng et al. *PNAS* 2017, 114, 1795). On CatalysisHub, we provide the atomistic structures used in our study for the readers to reference; the energetics calculated by others will reach similar conclusions. We believe our methods are fairly accurate and enable us to reach reasonable conclusions regarding *trends*. The use of DFT to model electrochemical interfaces inherently upholds certain assumptions, such as the large timescales used to find transition state complexes via NEB calculations or the tendency to delocalize charges across the periodic unit cell, or the electron densities are all ground-states in nature. Any future methodologies that subscribe to these assumptions will likely reproduce most of the trends found in this study.

DFT workflows typically do not change significantly as there is a need to stay consistent with past works. The meat of the work is whether it challenges previous conclusions and whether the work provides insights on how to address implications of the work.

We have added the following statement in our work (changes in blue font):

“In **Figure 2**, we will show that while our energetics align closely with previous works by Calle-Vallejo *et al.*²⁶ and Cheng *et al.*²⁹, our barriers reveal new implications regarding the formation of C₂ Oxy vs. HC on Cu surfaces.” (Page 5, Main Text)

We have also added **Table S3** to show that our reaction energetics relative to

gas-phase species are within reasonable accuracy to experimentally measured values (changes in blue font).

“To validate the afore obtained gas-phase energies, the energies at the standard condition (298 K, 101325 Pa) are used to calculate the equilibrium potentials for comparison with the experimental values obtained from the NIST Chemistry Webbook (<https://webbook.nist.gov/chemistry/>) and Langes Handbook of Chemistry. Most of the experimental data has been collected in the Supporting Information of Ref.⁹ The standard reaction enthalpies (ΔH^0) and equilibrium potentials (U_{eq}) are compared in **Table S3**. It is notable that the errors in ΔH^0 and U_{eq} are within $\pm 5 \text{ kJ mol}^{-1}$ and $\pm 0.02 \text{ V}$, respectively. Ethenone (CH_2CO) is an exception herein and the errors in ΔH^0 are calculated as 18.65 and $-20.55 \text{ kJ mol}^{-1}$, depending on the experimental data from different sources. The experimentally measured ΔH^0 varies by $\sim 40 \text{ kJ mol}^{-1}$. Overall, these results indicate that with the above DFT corrections applied to H_2 , CH_2CO , CH_3COOH , and CH_3CHO , reasonable agreement with thermochemical data is achieved.” (Page S3, SI)

The DFT calculations use a single adsorbate coverage for all adsorbates, though it is unknown to what extent coverage dependences will impact reaction energetics for all the steps involved, or whether the low coverage used represents the operating surface at all electrode potentials considered.

We thank the reviewer for their comment. Previous DFT studies show that CO is the major surface specie (Liu et al. Nat. Commun. 2019, 10, 32). We agree that the coverage effect on certain C_2 species may vary (such as revealed by Li et al. in Nat. Catal. 2019, 2, 1124; while this work considered different pathways that Cheng et al. proposed in PNAS 2017, 114, 1795). In fact, one of the recommended applications of our work is to study the coverage effect based on the key steps we found in a follow-up paper. And we have stated in **Supplementary Note 6** as

“Recently, the CO coverage effect has been demonstrated on tuning the C_2 Oxy/HC selectivity.⁴¹ Theoretical consideration of such a coverage effect on key C_2 intermediates such as CHCOH^* , OCHCH^* , and CH_2CO^* should be a future direction.” (Page S17, SI).

The cross-interaction parameters of key intermediates of CHCOH^* , CH_2CO^* , and CHCHO^* , as well as the corresponding transition states, to CO^* need to be explicitly obtained using the method described in Refs. 39 and 40 in SI (Lausche et al. J. Catal. 2013, 307, 275; Yang et al. J. Am. Chem. Soc. 2016, 138, 3705). Then these parameters could be directly used in our adsorbate-adsorbate interaction model shown in **Supplementary Note 6** (Pages S14–S17, SI).

The following changes were made to the manuscript method to state more clearly about how we consider the converge effect (changes in blue font):

“Based on previous works,^{25, 32} CO is the predominant intermediate on the Cu surface. We assume that there will be minimal coverage effects on reaction

barriers and energetics aside from interactions with CO, which have been included in the model.” (Page 21, Main Text)

Solvation and electrification effects are difficult to represent in electrochemical systems, and can have large effects reaction energetics that would alter conclusions. Solvation has been included in an entirely heuristic way with constant values roughly assumed for different adsorbates based on their structures.

We thank the reviewer for their comment. We agree that solvation effects are an open topic in electrochemical systems. Nevertheless, the correction scheme used in this study is reasonable compared to past works (Calle-Vallejo et al. *Angew. Chem. Int. Ed.* 2013, 52, 7282; Montoya et al. *J. Phys. Chem. Lett.* 2015, 6, 2032; Liu et al. *Nat. Commun.* 2019, 10, 32; Ludwig et al. *J. Phys. Chem. C* 2019, 123, 5999; etc.). Solvation effects in aqueous systems generally quantify the stabilizations of intermediates via H-bonding with surrounding water. Certain intermediates like H* and C* are assumed to be screened by the metal surface and are not likely to participate in H-bonding. We primarily analyze the solvation effects on adsorption energies of key intermediates of CHCO*, CH₂CO*, OCHCH* and CHCOH* using the model containing a metal slab and a layer of explicit water (please see **Supplementary Note 3** for details). The adsorption energies of these species at vacuum and solvation conditions are presented in the following table.

Table R1. Adsorption electronic energy in vacuum and solvation conditions

Specie	Adsorption energy (eV, vacuum)	Adsorption energy (eV, with explicit solvents)	Difference (eV)
CHCO*	-1.76	-1.87	-0.11
CH ₂ CO*	-2.00	-2.07	-0.07
OCHCH*	-2.39	-2.45	-0.06
CHCOH*	-1.68	-1.78	-0.10
CCH*	-1.41	-1.46	-0.05
CHCHOH*	-2.01	-2.12	-0.11

For simplicity, we used a constant -0.10 eV solvation correction for these species, and variations within a few tenths of meV (as shown in **Table R1**) will not impact the conclusions of this work. While for other species that we simply assume to have a similar correction, the minimum change in solvation correction required to alter our conclusion will be (0.78 eV - 0.40 eV) = 0.38 eV if we read the smallest difference in barrier heights from **Figure S6**. This corresponds to the selectivity determining protonation of the oxygen, or the primary (or secondary) carbon of the CHCOH* intermediate. However, we performed additional calculations using the explicit solvation method and could find in **Table R1** that the difference in solvation corrections between the two relevant species CCH* and CHCHOH* is still within a few tenths of meV, further supporting our assumption and conclusions.

We have added the following statement to the manuscript (changes in blue font):

“Although strong solvation effects on OH* adsorption have been widely discussed in literature,^{51, 52} it is not currently possible to judge if the value chosen for ΔG_{OH^*} in this work is corrected or not. However, significant deviations will not affect the conclusion of this work because ΔG_{OH^*} only serves as a descriptor; OH solvation corrections would merely lead to a constant shift for the y-axis in the selectivity map of **Figure 5** while not changing the relative positions of and the trends among different materials. In addition, exchange-and-correlation functional that accounts for long-range interactions, such as BEEF-vdW, generally results in a decrease in the strength of solvation contributions to the adsorption energies, providing a possible explanation to the relatively small solvation corrections we used in this work.^{53, 54}” (Pages 19–20, Main Text)

“Three-layer (4×3) supercells with a monolayer of explicit water, *i.e.* the same setting for electrochemical barrier calculations, were employed to investigate the solvation effect on several key intermediates identified in this work, including CH₂CO*, OCHCH*, CHCOH*, CHCHOH*, CCH*, CH₂CHOH*, and OCHCH₃*. Except CH₂CHOH* and OCHCH₃*, the studied five important C₂ intermediates have a similar $E_{\text{solv corr}}$ of around -0.10 eV, while CH₂CHOH* and OCHCH₃* have a distinct $E_{\text{solv corr}}$ of -0.51 eV and -0.25 eV, respectively. Due to the similar binding geometry, OCH₂CH₃* was assumed to have the same $E_{\text{solv corr}}$ of -0.25 eV as OCHCH₃*. The rest of C₂ species were assumed to have a $E_{\text{solv corr}}$ of -0.10 eV according to the results obtained for CH₂CO*, OCHCH*, CHCOH*, CHCHOH*, and CCH*.” (Page S5, SI)

Electrification assumes an arbitrary Helmholtz model, does not consider PZC effects of reaction conditions, and makes an arbitrary (and quantitatively important) choice for the distance of charge separation.

We thank the reviewer for this insightful comment. While the Helmholtz model used in our study does utilize some arbitrary parameters, it works well for reproducing energetics of the intermediates that are most sensitive to electrification like OCCO* (for the comparison with literature, please refer to Figure S3 of our previous work Energy Environ. Sci. 2021, 14, 473) and CO₂*. In our study, we used a constant PZC for simplicity, the value of which was directly adopted from literature (Trasatti et al. Modern aspects of electrochemistry, Springer, 2002; Ringe et al. Energy Environ. Sci., 2019, 12, 3001). Although PZC could vary with CO* adsorption effect, we performed a sensitivity analysis to show that neither tuning the PZC nor varying the distance of charge separation impact our conclusions (Figure S11). The selected ranges of d and U_{PZC} for such sensitivity analyses are based on the effective interfacial ion radii and experimental U_{PZC} for Cu(100), Cu(111), and polycrystalline Cu according to Ringe et al. (Energy Environ. Sci., 2019, 12, 3001). **Figure S11** is shown below:

Figure S11. Sensitivity analyses with varying U_{PZC} and d . The effect of d on (a) TS energies of CHCO-H, OCCH-H, and CHC-HO steps and (b) C_2 Oxy/HC ratio. The colors change from dark to light as d increases. The bold lines refer to the case of $d = 1.2 \text{ \AA}$, which is used throughout the other part of this work. For a and b, U_{PZC} equals to -0.54 V_{SHE} (for Cu(100)). The effect of U_{PZC} on (c) TS energies of CHCO-H, OCCH-H, and CHC-HO steps and (d) C_2 Oxy/HC ratio. The solid lines refer to the case of $U_{PZC} = -0.54 \text{ V}_{SHE}$, which is used throughout the other part of this work. For c and d, d equals to 1.2 \AA .

We have added the following to the manuscript to clarify the minor effect of the selection of the PZC (U_{PZC}) and charge-separation distance (d) parameters in our Helmholtz model on the conclusion (changes in blue font).

“A previous study by Ringe *et al.* has suggested that both the ion identity and the CO adsorption could affect the interfacial field effect on CO_2R .¹⁸ Therefore, the selection of parameters in our electric field model, i.e., the zero-charge potential (U_{PZC}) and charge-separation distance d (see **Supplementary Note 4** for more details) might have an effect on our results. To investigate that, we performed sensitivity analyses of both the key energies (TS energies of the three SDSs) and the C_2 Oxy/HC ratio by varying U_{PZC} and d within reasonable ranges. The value of d varies from 1.2 \AA to 2.0 , 3.0 , 4.0 , 5.0 , and 6.0 \AA . The 4.0 , 5.0 , and 6.0 \AA approximate the effective interfacial cation radius of K^+ , Na^+ , and Li^+ , respectively. The value of U_{PZC} varies from -0.54 V_{SHE} (for Cu(100)), to -0.20 V_{SHE} (for Cu(111)), to 0.09 V_{SHE} (for pcCu). All these values were directly adopted from Ref.¹⁸. The influence of varying U_{PZC} and d is shown in **Figure S11**. Although

changing these parameters in our electric field model has an effect on the TS energies, the energetic difference, e.g., $\Delta G_{a,0}^{\text{CHCO-H}} - \Delta G_{a,0}^{\text{OCCH-H}}$, is hardly influenced (Figures S11a, c). Thus the overall trend in C₂ Oxy/HC ratio is not altered (Figures S11b, d). Note that the right leg in Figure S11d suffer from larger deviations. This suggests a potential strategy to further engineer the C₂ Oxy/HC selectivity at low overpotentials via the U_{PZC} effect.” (Page S36, SI)

The approach to calculate barriers suffers from similar assumptions/approximations, and does not successfully establish whether the initial state to which barriers are referenced properly captures the chemical potential of the bulk phase ion reactant. The assumptions made as to how energetics are altered whether the proton donor is a hydronium ion or water molecule essentially “bake in” the conclusions as to pH effects rather than allowed the model to analyze these effects.

It is important that this aspect comes across clearly, and we appreciate the comment of the reviewer. We use the computational hydrogen electrode model to determine the energy of protons relative to the bulk solution for electrochemical reactions. We believe there has been a misunderstanding between us and the reviewer regarding how states are referenced/used to calculate barriers; we have used the final state of each reduction reaction to determine barriers. We only use the NEB images to capture the structure of the transition state complex and the charge it contains. The charge extrapolation scheme is then used to obtain a backward barrier relative to the final state; the forward barrier is then derived based on the backward barrier and the reaction energetics. We have revised the text to make sure this is clear for our readers.

We also want to emphasized that our work does not `explained away` pH effects via changes in proton source. Throughout the study, the proton source remains unchanged as water because we primarily focused on neutral and alkaline conditions that are relevant to most of the experiments. However, it is a common computational trick to use the hydronium ion ($[\text{H}_3\text{O}]^+$) as proton donor in the models as this, in contrast to H₂O, can be correctly simulated in proton-coupled electron transfer reactions. The conversion to H₂O as donor is achieved through a constant, well-established empirical energy shift. In the **Supplementary Note 5** and **Figure S2**, we have shown how ΔpH affects reaction energetics and thus would affect the barriers for certain steps over others.

We have added the following text in the SI (changes in blue font).

“Recent experimental and theoretical works also showed that water should be the dominant hydrogen source for PET in CO₂R and HER at neutral and alkaline conditions.^{29, 30} However, currently it is not trivial to model alkaline barriers with water using DFT.^{12, 30} Thus, we use hydronium as a proton source with an alkaline correction applied based on other works.^{5, 12, 30-32} The following are detailed analysis regarding pH effects on the IS/FS energies and the choice of potential

scales of RHE/SHE.” (Page S10, SI)

Though all of these limitations are inherent to the established approaches used in this study to model electrocatalytic energetics, when even further coupled with inherent DFT error (and lack of consideration of interfacial entropic effects), the methods used are collectively not precise enough to answer the research questions posed. The methods used have been competently executed, but simply suffer too substantially from the inherent difficulties in DFT representation of electrocatalytic energetics to reaction definitive and substantive conclusions. Further, much of these approximations are only apparent to the expert (and mainly noted on a detailed read of the SI), and the manuscript itself does not provide a sufficient quantification or critical discussion of these limitations.

We thank the reviewer for the comment. The methods used in this study are based on first-principles: energetics of intermediates are calculated on slabs without water structures, which allow us to calculate deterministic and reproducible reaction energies between intermediates; NEBs are used to capture the structure and partial charge of the transition state; barriers are then referenced from the final state of each intermediate and pH/solvation effects are applied within reasonable accuracy compared with previously reported works (Nie et al. *Angew. Chem. Intl. Ed.* 2013, 52, 2459; Cheng et al. *PNAS* 2017, 114, 1795; Liu et al. *Nat. Commun.* 2019, 10, 32; Patel et al. *J. Phys. Chem. Lett.* 2021, 12, 5193; etc.). While we agree with the reviewer that electrochemical interfaces are complex and that more attention should be made to incorporate the plethora of effects that encompass electrochemical reaction energetics, our work provides initial trends and reachable conclusions despite shortcomings in DFT. In order to thoroughly model proton-coupled electron transfer in alkaline conditions, we detailed our methodology in **Supplementary Note 5** (Pages S10–S16, SI). We strive to make our methods and approximations as clear to the reader as possible without having the paper become too lengthy (hence most of the computational detail are placed in the SI). We further discuss the limitations of our methods below and ways to address them:

The 0.1 to 0.2 eV inherent error in GGA-level DFT when benchmarked with experimental data generally comes from poor energetic quantification of certain chemical bonds; for gas-phase molecules containing C=O double bonds (CH₂CO, CH₃COOH, and CH₃CHO), a DFT correction of +0.15 eV was added according to Christensen et al. (*Catal. Sci. Technol.* 2015, 5, 4946), and a DFT correction of +0.09 eV was added for hydrogen gas due to the systematic error caused by the BEEF-vdW functional, in accordance with Studt et al. (*ChemCatChem* 2015, 7, 1105) (Page S2, SI). We have added a new table in the SI (**Table S3**) that compares thermochemical energetics between experiment and DFT to show that we strive to stay within reasonable accuracy to experimentally attained energetics.

Solvation effects are explicitly calculated for certain intermediates and compared with

those found in literature. Aside from OH*, these correction schemes are minor and do not affect the overall conclusions of the paper.

Despite our naïve Helmholtz model, we show through sensitivity analysis that neither tuning the PZC nor varying the distance of charge separation impact our conclusions (Figure S11) via electrification effects.

We have now quantified and discussed the limitations/assumptions of our methodology and summarized them in **Supplementary Note 9**. In the main text, we also added the followings to remind readers about these limitations:

“Since challenges remain for modeling electrochemical activation barriers, the underlying limitations/assumptions from our methodology are listed in **Supplementary Note 9**.” (Page 21, Main Text)

The limitations/assumptions of our methodology were listed as:

“Put succinctly, the underlying limitations/assumptions from our methodology that we cannot fully address are as followed:

- Our TS complexes are found with CI-NEB, which allow water molecules ample time to relax to their ground state even though PET steps are measured to be very fast.
- We assume that the first water layer is arranged in an ice-like manner. The water layer can be obtained via minima-hopping. Under reducing conditions, the water molecules face down with hydrogens pointed towards the surface.
- The CHE model inherently assumes electron transfer is in sync with each other; we do not consider steps like one-electron-two-proton transfer. A notable limitation to this assumption is the inability to model intermediates with a charged state such as a CO_2^- .^{47, 48}
- We assume Cu is not in a positive oxidation state.
- We assume cations and water molecules do not adsorb on the surface.
- We assume that the TS complex found via protonation using H_3O^+ is similar to that with protonation using H_2O , such that a constant correction scheme can be used as an extrapolation.
- We assume that an electrochemical interface can be modeled as a plate capacitor. This allows us to separate the energy contribution from chemical bonding and electrostatics.
- We assume that electrification and solvation effects are entirely independent from each other; the calculations of electrification only model the intermediates on a bare slab.
- Under Bader charge partitioning of the TS complex, the charge of the transferring proton is *not* considered part of the TS complex. This is despite the fact that we consider the transferred proton in the TS complex when computing vibrational modes.

We believe that as long as future works subscribe to these assumptions, the findings

of this work are upheld.⁴⁹ We note that these assumptions have been largely successful attaining microkinetic models that can model the product rates and distributions on Cu.^{5, 12, 28} Competing mechanistic models that pick apart these assumptions will also need to show their capabilities to account for product rates and distributions across pH and applied potentials.” (Page S21, SI)

Regarding possible future work on improving the accuracy of DFT methods, our group has recently developed a method to model potential-dependent electrochemical activation barriers, which explicitly include charge conservation, different time scales of involved processes, and the thermal fluctuation of water at 298 K (Li et al. J. Am Chem. Soc. 10.1021/jacs.1c07276). This new method was proposed using alkaline hydrogen evolution reaction on Pt(100) as a benchmark reaction; but this generic method is fully transferrable to CO₂R as it only requires an explicit evaluation of decoupled reactant and product energy profiles and simulated solvent fluctuations at the interface under relevant operating conditions. Revisiting the selectivity-determining steps we identified in this work using this newly developed method for modeling electrochemical activation barriers is definitely one of the future works worth being investigated.

We have added the followings to the manuscript (changes in blue font):

“In addition, we would like to clarify herein that while accurate theoretical simulation of electrochemical activation barriers is inherently difficult using DFT-based approaches, there are new methods recently developed to capture the general trends in protonation barriers,⁴⁹ or to explicitly include thermal fluctuations of the solvent at the electrochemical interface.⁵⁰ Revisiting the SDSs we identified in this work using these methods is definitely one of the future works worth being investigated.” (Page S21, SI)

2) The electroreduction of CO or CO₂ on Cu electrodes, empirically, does not provide sufficient selectivity to any C₂ product to be of any practical use. The work done here is only directly relevant to a single electrode surface, for which numerous DFT studies already exist, and makes the relatively minor advance of considering more details of specific C₂ product-determining steps. Given the lack of practical relevance of electrode studied, and incremental nature of the mechanistic advances, the impact of further detailed DFT work on CO/CO₂ reduction on Cu is minimal. Frankly, there seems to be a self-perpetuating cycle of publications in this area without any significant practical advance beyond well established knowledge on the performance of Cu electrodes for CO₂ electroreduction.

We thank the reviewer for the comment. In order for CO₂R on Cu to scale for industrial applications, we need to address the question whether it is possible to tune the selectivity towards a particular C₂ product. To answer such a question, we must consider not only the thermodynamics of the mechanistic pathways (which most of the previous mechanistic studies have *only* provided), but also the *kinetics* of the

intermediate steps. This is what makes this study special. It provides the most comprehensive dataset as to why selectivity towards C₂ product is poor and exposes the immense challenge on finding surfaces that can sway selectivity towards either ethylene or ethanol. We sympathize with the reviewer that simply examining Cu `again` seems futile and self-perpetuating, but it is currently the most practical material that can achieve multi-carbon product distributions for CO₂R. Any new material discovery computational project considered for CO₂R will still need to consider all the effects and concerns that the reviewer have raised (electrification effects, solvation effects, pH effects, surface reconstruction effects, coverage effects, mass transfer limitations, thermodynamic stability, barrier accuracy, commercial cost). It is possible to expand towards this direction, but significant resources will be needed to address this demand. This study provides some motivation for whether new materials are worth considering and also serves as a neat framework for how CO₂R mechanistic studies can be carefully carried out.

We will improve our motivation in the introduction to highlight that our work addresses the intrinsic selectivity problem of CO₂R to make ethanol or ethylene. The following changes have been made in the introduction of the manuscript (changes in blue font):

“Among possible C₂ products, C₂ oxygenates (Oxys) such as ethanol and acetic acid are produced in liquid forms and possess high volumetric energy density and are compatible with existing infrastructure for easy storage and transportation.^{1, 3} The majority C₂ hydrocarbon (HC) product on Cu is ethylene, which is also a widely used chemical feedstock in industrial processes.^{3, 4} Despite the ability to produce C₂₊ products, pristine Cu is not selective towards a specific C₂₊ product. Thus, the fundamental selectivity issues of CO₂R on Cu is a challenge of great importance in catalysis and sustainable energy technologies. Substantial focus has been made in enhancing the selectivity of C₂ products for CO₂R.⁵⁻¹¹” (Page 2, Main Text)

Reviewer #3 (Remarks to the Author):

Electrocatalytic CO and CO₂ reduction is a hot topic; there are every month new papers in high impact journals on this. The interest is twofold: firstly, it would be a breakthrough in power-to-x technology if these reactions are catalyzed in a selective and efficient manner, secondly, the chemistry on a Cu surface is very rich and complex, which means that it is a challenge to obtain fundamental insight. So far, nobody have found a great novel catalyst for this reaction. Therefore, most papers, including this one, address the understanding of the reactions. The carbon-carbon bond that can form on Cu is a puzzle and understanding the selectivity is from a fundamental standpoint very interesting.

The amount of work behind this paper is huge and the analysis is careful. In the analysis of electrochemical reactions there are a number of choices done to setup the model and reaction path. However, the results are very sensitive to the exact manner in which the model constructed. This means that it is difficult obtain conclusive insights.

We thank the reviewer for this comment. We agree that the approximations used in this study are important for constructing the model; modeling electrochemical interfaces evidently requires many parameters. Many of these approximations are consistent with past studies and should not affect the conclusion of the paper. We also appended additional sensitivity analyses and found little influence on our conclusion (**Figure S11**).

Figure S11. Sensitivity analyses with varying U_{PZC} and d . The effect of d on (a) TS energies of CHCO-H, OCCH-H, and CHC-HO steps and (b) C_2 Oxy/HC ratio. The colors change from dark to light as d increases. The bold lines refer to the case of $d = 1.2 \text{ \AA}$, which is used throughout the other part of this work. For a and b, U_{PZC} equals to -0.54 V_{SHE} (for Cu(100)). The effect of U_{PZC} on (c) TS energies of CHCO-H, OCCH-H, and CHC-HO steps and (d) C_2 Oxy/HC ratio. The solid lines refer to the case of $U_{PZC} = -0.54 \text{ V}_{SHE}$, which is used throughout the other part of this work. For c and d, d equals to 1.2 \AA .

This is reflected in Fig 5. Cu 211, 511, 310 are very different in measured selectivity, however, in OH and C binding they are very close to each other. This probably means that there is something, not considered in the model, responsible for the differences in experiments for those facets. I would like the author to comment on that.

We thank the reviewer for the comment. We agree that those Cu step/kink surfaces have similar OH* and C* binding energies. We will clarify that the points plotted on Figure 5 are only calculated binding energies from the most reactive site on the surface (the site around the step). Experimentally, surfaces may reconstruct and their structures may deviate from the ideal surface in which computations model with. The threshold boundaries depicted in white are not hard boundaries, but areas of uncertainty. Nevertheless, the merit of the OH* and C* descriptor model is that it uses mechanistic understandings of this work to extrapolate/predict what the C_2 products would be for other materials should CCO* become possible on those respective surfaces. For instance, if Au/Ag were to form C_2 products, it would mainly form ethanol based on our descriptor model. And this is also consistent with the experimental observations that decorating Cu with Ag generally resulted in enhanced ethanol or other C_2 Oxy vs. ethylene.

We have added a statement stressing the limitations of the model within the decision boundaries of Figure 5 (changes in blue font).

“One limitation to the selectivity map in **Figure 5** is that the decision boundaries are not hard boundaries. Most of the Cu facets are located close to the boundaries between the C_2 HC- and the C_2 Oxy-selective regions, highlighting their ability to produce a variety of C_{2+} products, but also the immense challenge of engineering Cu catalysts for $CO_{(2)}R$ with a single product selectivity.” (Page 16, Main Text)

I also think the authors could be more clear in their claims, however, the paper has a high quality due to the amount of data and the careful treatment. I can therefore recommend that the paper is accepted in Nature Com.

We thank the reviewer for the recommendation. The valuable suggestions have enabled us to further improve our work.

Reviewer #1 (Remarks to the Author):

The authors addressed most of my comments. I am still concerned about the small solvation corrections found/used in this work, but at least the authors provide the values and make the readers aware of the problem. Thus, I recommend this paper for publication in Nature Communications.

Reviewer #2 (Remarks to the Author):

The reviewers have supplemented a detailed work with conscientious and well developed responses to all reviewer's comments. Revisions made to the paper help highlight both the contributions and the limitations of the work. I remain convinced that the inability to quantify the impacts of coverage, solvation, and electrification, together with the imprecision inherent in choice of XC functional, make the methods used not capable of precisely answering the mechanistic questions regarding the CO₂/CO electrolysis product distribution. The authors defend well their approach relative to other computational literature, all of which also suffers from the same inherent challenges. Despite these limitations, this paper is certainly on the top end of DFT examinations of this reaction chemistry. It will certainly receive the attention of being cited given the population of researchers still studying CO₂ reduction on Cu.

Reviewer #3 (Remarks to the Author):

I find that the authors have answered my comments. I did already recommend accepting the manuscript.

Reviewer #1 (Remarks to the Author):

The authors addressed most of my comments. I am still concerned about the small solvation corrections found/used in this work, but at least the authors provide the values and make the readers aware of the problem. Thus, I recommend this paper for publication in Nature Communications.

We thank the reviewer for a careful review and insightful questions. We will continue to follow the rich literature on solvation models in computational chemistry and ensure to adapt to the state-of-the-art also in our future work.

Reviewer #2 (Remarks to the Author):

The reviewers have supplemented a detailed work with conscientious and well developed responses to all reviewer's comments. Revisions made to the paper help highlight both the contributions and the limitations of the work. I remain convinced that the inability to quantify the impacts of coverage, solvation, and electrification, together with the imprecision inherent in choice of XC functional, make the methods used not capable of precisely answering the mechanistic questions regarding the CO₂/CO electrolysis product distribution. The authors defend well their approach relative to other computational literature, all of which also suffers from the same inherent challenges. Despite these limitations, this paper is certainly on the top end of DFT examinations of this reaction chemistry. It will certainly receive the attention of being cited given the population of researchers still studying CO₂ reduction on Cu.

We appreciate that the reviewer is challenging the employed modeling methods as it is important to ensure that we as a community continue to evolve the field and that new studies adapt to the current state-of-the-art. We do agree with the reviewer that there, in general, is room for improvement in certain aspects of DFT modeling, not least in the modeling of electrochemistry. The comments from the reviewer have helped nuance the discussion in the paper and have improved its quality. We are indeed very thankful for the input.

Reviewer #3 (Remarks to the Author):

I find that the authors have answered my comments. I did already recommend accepting the manuscript.

We thank the reviewer for valuable feedback that has helped making the manuscript stronger, and more accessible to the readers.